# Impact of measured and simulated tundra snowpack properties on heat transfer

Victoria R. Dutch[1], Nick Rutter[1], Leanne Wake[1], Melody Sandells[1], Chris Derksen[2], Branden Walker[3], Gabriel Hould Gosselin[4], Oliver Sonnentag[4], Richard Essery[5], Richard Kelly[6], Phillip Marsh[3], Joshua King[2], Julia Boike[7,8]

[1] Department of Geography and Environmental Sciences, Northumbria University, Newcastle upon Tyne, UK
[2] Climate Research Division, Environment and Climate Change Canada, Toronto, Canada
[3] Cold Regions Research Centre, Wilfrid Laurier University, Waterloo, Canada
[4] Département de géographie, Université de Montréal, Canada
[5] School of Geosciences, University of Edinburgh, UK
[6] Department of Geography and Environmental Management, University of Waterloo, Canada
[7] Alfred Wegener Institute, Helmholtz Centre for Polar and Marine Research, Potsdam, Germany
[8] Geography Department, Humboldt-Universität zu Berlin, Germany

*Correspondence to*: Victoria Dutch (victoria.dutch@northumbria.ac.uk)

**Abstract.**

Snowpack microstructure controls the transfer of heat to, and the temperature of, the underlying soils. In situ measurements of snow and soil properties from four field campaigns during two winters (March and November 2018, January and March 2019) were compared to an ensemble of CLM5.0 (Community Land Model) simulations, at Trail Valley Creek, Northwest Territories, Canada. Snow MicroPenetrometer profiles allowed snowpack density and thermal conductivity to be derived at higher vertical resolution (1.25 mm) and a larger sample size (n = 1050) compared to traditional snowpit observations (3 cm vertical resolution; n = 115). Comparing measurements with simulations shows CLM overestimated snow thermal conductivity by a factor of 3, leading to a cold bias in wintertime soil temperatures (RMSE = 5.8 °C). Two different approaches were taken to reduce this bias: alternative parameterisations of snow thermal conductivity and the application of a correction factor. All the evaluated parameterisations of snow thermal conductivity improved simulations of wintertime soil temperatures, with that of Sturm et al. (1997) having the greatest impact (RMSE = 2.5 °C). The required correction factor is strongly related to snow depth ($R^2 = 0.77$, RMSE = 0.066) and thus differs between the two snow seasons, limiting the applicability of such an approach. Improving simulated snow properties and the corresponding heat flux is important, as wintertime soil temperatures are an important control on subnivean soil respiration, and hence impact Arctic winter carbon fluxes and budgets.

## 1 Introduction

Seasonal snow is an effective insulator, with snow thermal properties influencing the soil microclimate (Lawrence and Slater, 2009; Wilson et al., 2020) and the distribution and state of permafrost (Biskaborn et al., 2019; Goncharova et al., 2019; Zhang, 2005). The temperature of the subnivean environment, particularly the extent to which it allows for the presence of small amounts of liquid water, acts as an important control on biogeochemical cycling, including soil respiration (Semenchuk et al., 2015; Sullivan et al., 2008; Williams et al., 2009). In addition, the soil temperature also impacts hydrology through controls on soil infiltration and runoff (Niu and Yang, 2006; Quinton and Marsh, 1999). Accounting for how well the thermal and hydrological conditions of subnivean soils (including the physical state of soil water content) are simulated is therefore critical for understanding how well current land models such as the Community Land Model (CLM; Lawrence et al. (2019)) simulate winter carbon fluxes (e.g. Natali et al. (2019)) and permafrost evolution (Koven et al., 2012).

The depth, [micro]structure, and stratigraphy of a snowpack determine its capacity to insulate the underlying soil and are in turn influenced by the temperature of the ground surface. Tundra snowpacks typically consist of a basal depth hoar layer, formed as strong temperature gradients within the snowpack induce kinetic metamorphism, overlain by an upper wind slab

layer, compacted and densified over the course of a snow season by strong Arctic winds (Sturm et al. (1995), Derksen et al,
(2009; 2014); Rees et al. (2014), among others). Between these two layers, an indurated hoar layer may also be formed (Sturm
et al., 2008), where the lower part of the wind slab takes on some of the microstructural properties of depth hoar (e.g faceted
grains) while maintaining the density and hardness of a wind slab (Derksen et al., 2009).

The thermal influence of the snowpack on the underlying soil can be considered in terms of an effective snow depth ($S_{depth,eff}$),
which describes the insulative properties of the snowpack by weighting the mean monthly snow depth by its relative position
in the season at a given location across an entire winter (October – March) (Slater et al., 2017), emphasizing the timing of
snow accumulation as more important than the end of season snow depth in determining wintertime soil temperatures
(Lafrenière et al., 2013). Rapid snow accumulation and snowpack establishment early in the winter will insulate the ground
thereby dampening soil temperature fluctuations, leading to a higher $S_{depth,eff}$ than steady accumulation throughout the entire
winter, even if the total amount of precipitation is the same (Slater et al., 2017). The relationship between $S_{depth,eff}$ and the
normalised temperature difference between air and soil ($A_{norm}$) can be used to understand heat transfer between the air and the
soil and through the snowpack (Slater et al., 2017). The deviation of this relationship from the expected exponential form
(Slater et al. (2017) - Fig. 3), termed the Snow Heat Transfer Metric (SHTM), can be calculated and used to evaluate simulated
heat transfer processes in the soil and snowpack as was undertaken by Slater et al. (2017) for the land surface components of
participating models in the CMIP5 model intercomparison project (Taylor et al., 2012). The closer the value of the SHTM is
to one, the smaller the disagreement between modelled and observed air and soil temperature differences. Being able to
quantitatively assess snow heat transfer is of particular importance because model parameterisations of snow physical
properties can lead to differences in soil temperature and therefore contribute to uncertainties in estimates of Arctic winter
carbon fluxes and budgets, which are currently not well constrained (Fisher et al., 2014; Natali et al., 2019; Virkkala et al.,
2021).

The effective thermal conductivity of the snowpack ($K_{eff}$; heat conducted through ice and interstitial air) determines the rate
of heat transfer to underlying soil (Domine et al., 2015; Jafarov et al., 2014). From here on, we refer to the effective thermal
conductivity of the snowpack as snow thermal conductivity for brevity, after Jafarov et al. (2014). Snow has a low thermal
conductivity, typically in the range $0.01 – 0.7$ $Wm^{-1}$ $K^{-1}$ (Gouttevin et al., 2018). Typical $K_{eff}$ values for tundra snowpacks are
at the lower end of this range, for example Domine et al. (2016) found a maximum value of 0.33 $Wm^{-2}$ $K^{-1}$. Measurement of
snow thermal conductivity is typically undertaken using a heated needle probe (Morin et al., 2010), although snow anisotropy
causes 29 % uncertainty in these estimates of $K_{eff}$ (Domine et al., 2015), which is a notable limitation to this method (Riche
and Schneebeli, 2013). Models typically parameterise $K_{eff}$ as a function of the simulated snow density (Gouttevin et al., 2018),
for which a number of different statistical relationships have been proposed (e.g. Sturm et al. (1997); Calonne et al. (2011)).

This study characterises the variability of the thermal properties of tundra snow and resultant soil temperatures at Trail Valley
Creek, Northwest Territories, Canada, over the 2017 - 18 and 2018 - 19 winters using in situ measurements. We then use these
measurements to evaluate an ensemble of simulations from the Community Land Model (CLM5.0), particularly with regard
to how thermal properties are simulated and the sensitivity of soil temperatures and SHTM to the properties of the snowpack.

## 2 Data and methods

### 2.1 Study location

Trail Valley Creek (TVC; 68°45'N, 133°30'W) is a 57 $km^2$ boreal-tundra transition research watershed located in the Inuvialuit
Settlement Region, approximately 55 km northeast of Inuvik, NWT, Canada. TVC has an average elevation of approximately
99 m above sea level (Marsh et al., 2008) and a mean annual air temperature of -7.9 °C for the period 1999 - 2018 (Grünberg
et al., 2020). Land cover at TVC predominately consists of graminoid tundra, with some lakes, small clusters of willow and
alder shrubs and some isolated black spruce stands (Essery and Pomeroy, 2004; Grünberg et al., 2020; King et al., 2018). The

terrain consists of mineral soil hummocks of up to a metre in diameter, and peaty inter-hummock hollows (Quinton and Marsh, 1998). The ground is underlain by continuous permafrost to a depth of 350 - 500 m (Wilcox et al., 2019), with a maximum active layer depth of up to 1 m at the end of the summer (Grünberg et al., 2020). Snow cover at TVC has a typical duration of 8 months (Pomeroy et al., 1993), with typical depths of 0.2 - 0.5 m, though drifts exceeding 1-2m occur surrounding tall shrubs and in proximity to steep slopes (Marsh and Pomeroy, 1999).

## 2.2 Field methods

Comprehensive snow and soil data are used from four winter season intensive measurement periods (14 - 21 March 2018; 12 - 18 November 2018, 11 – 20 January 2019, and 18 – 27 March 2019). Additionally, meteorological data for the entirety of the study period (1 August 2017 – 31 August 2019; plus model spin-up), measured at the TVC eddy covariance tower (AWS) were also used. Half-hourly 2 m air temperatures were measured using a HMP35CF sensor (Campbell Scientific, Logan, Utah) and precipitation totals were measured using a weighted T-200B gauge (Geonor Inc., Branchville, New Jersey). Precipitation gauge under-catch is common in tundra environments such as TVC (Smith, 2008; Watson et al., 2008; Gray and Male, 1981), therefore precipitation was corrected as per Pan et al. (2016). Automated snow depth measurements used were from the nearby Meteorological Service of Canada station and measured by a SR50a sensor (Campbell Scientific). Soil temperature profiles (Boike et al., 2020) were measured at 2, 5, 10 and 20 cm depths using 107B Thermistors (Campbell Scientific). Soil moisture content (Boike et al., 2020) was profiled at the same depths using CS615 soil water content reflectometers (Campbell Scientific).

Spatially distributed Snow MicroPenetrometer (SMP; Schneebeli and Johnson (1998)) profiles (n = 1050) were measured across the TVC sub-catchment. The SMP provides vertical profiles of force at 40 µm resolution (Proksch et al., 2015). Bespoke coefficients for tundra snowpacks were calculated based on the methodology of King et al. (2020b) to derive high vertical resolution snow density profiles from the SMP force profiles (see Appendix A for detailed methodology). Briefly, a K-folds recalibration was used to derive new coefficients (Table A1) from 36 co-located snowpits and SMP profiles across the TVC catchment. These coefficients were then applied to all 1050 SMP force profiles from the 3 campaigns over a 2.5mm rolling window to give recalibrated density profiles. These density profiles were then used to approximate profiles of thermal conductivity using the $K_{eff}$ relationships derived by Sturm et al. (1997), Calonne et al. (2011), Jordan (1991), and Fourteau et al. (2021b), denoted $K_{eff-Sturm}$, $K_{eff-Calonne}$, $K_{eff-Jordan}$ and $K_{eff-Fourteau}$ respectively. Use of the SMP allows for a large increase in both the number of sites and the vertical resolution at each site compared to traditional snowpits, but some coincident snowpit measurements are still required to derive the coefficients to estimate snow density. Sources of uncertainty in the SMP measurements include interactions with vegetation within the snowpack and collapse of the depth hoar layer during measurement; an experienced SMP user can easily identify and remove profiles which are affected by these issues. A positive bias in derived depth hoar density occurs because of large distances between snow grain failures (see Appendix A and King et al., 2020b for more details).

During the March 2018 and March 2019 campaigns, thermal conductivity was also measured using a TP02 needle probe (Hukseflux, Delft, Netherlands) after Morin et al. (2010). Measurements of thermal conductivity of each snowpack layer, a total of 105 measurements from 37 different snowpits were made across these two campaigns. Almost 36,5000 GPS located snow depths (Toose et al., 2020; King et al., 2020a) were measured across the 4 campaigns using a Magnaprobe instrument (Sturm and Holmgren, 2018), allowing spatial distributions of snow depths across the catchment to be examined. Vertical profiles of snow density, using a 100 $cm^3$ box cutter (Conger and Mcclung, 2009), and snowpack temperature were measured at all snowpit locations for each campaign. Stratigraphic information profiled in each snowpit (n = 115) was used to assign one of four different layer types (surface snow, wind slab, indurated hoar and depth hoar) to the measured densities (Fierz et al., 2009) in order to assess spatial variability in the thickness and properties of different snowpack layers.

## 2.3 Snowpack simulations

The Community Land Model v5.0 (CLM; Lawrence et al. (2019)) is the land surface component of the Community Earth System Model v2.0, which can be run at a variety of spatial scales. In this study, 1D "point mode" (a 0.1° x 0.1° grid cell) CLM (PTCLM; Kluzek (2013)) simulations were centred at the location of the TVC station. Minor adjustments were made to the model in order to better emulate snow accumulation and melt at the point scale; the snow accumulation factor was increased (Swenson and Lawrence, 2012) from 0.1 to 2.0 and the standard deviation of elevation set to 0.5 m after Malle et al. (2021; Figure S4). These adjustments limit the period of fractional snow cover, so that PTCLM represents a binary state of snow presence or absence over a flat surface. PTCLM simulations were run from August 2017 to August 2019, with model spin-up from January 2013. Spin-up of PTCLM was necessary in order to allow soil temperatures to equilibrate. Variation between model runs with the same parameterisation after more than 2 full years of spin-up is limited to ~ 1 °C throughout the top 5 m of the soil column. The impact of spin-up on soil temperature is further discussed in Appendix B.

Simulations were forced with gap-filled AWS data from TVC. Following Essery et al. (2016), gaps of 4 hours or less were filled using linear interpolation and larger gaps filled using ERA5 reanalysis data (Hersbach et al., 2020). Gapfilling was only required for measurements of incoming longwave and shortwave radiation, and comparison of observations and reanalysis data showed an offset of less than 60W m$^{-2}$. Bias correction of reanalysis data was not undertaken due the small size of this offset. Daily precipitation amounts from the AWS were converted to the hourly resolution required by CLM using the fraction of daily precipitation at each hourly timestep from ERA5. ERA5 reanalysis data was also used to partition precipitation into rain and snow for comparison against the linear ramp used by CLM. All precipitation falling when air temperatures are below 0 °C is classed as snow, after which point an increasing proportion of the precipitation is classed as rain until air temperatures are above 2 °C where all precipitation is classed as rain (Lawrence et al., 2019).

Developments between CLM4.5 and CLM5.0, as outlined in Van Kampenhout et al. (2017) improved the snow scheme in CLM. The version of the model used herein produces a computationally-layered snowpack, with the number of snow layers dependant on the snowpack depth, up to a theoretical maximum of 12 layers (as opposed to the 5 layer maximum in previous versions of CLM). Once the total snow depth exceeds a given threshold, the initial snow layer is subdivided into two layers with equal properties. Snow layer formation continues in this manner as layer thicknesses surpass the prescribed ranges given in Jordan (1991). When a layer divides, the new layer is formed beneath it, rather than new layers being formed at the surface by new snowfall. As this process is not stratigraphically representative, layers are not described by snow type (for example, as per Fierz et al. (2009)), but instead numbered from the snow surface down. Layer thicknesses are also influenced by snow compaction, parameterised following Anderson (1976). Unsaturated layers may compact due to overburden pressure, the breakdown of new snow crystals or melting, with the thickness of a snow layer a function of the snow thickness at the previous timestep and the rate of compaction. Snow depths below 1 cm are not discretely modelled and are instead combined into the surface soil layer.

Density, thickness and thermal conductivity are output as a daily mean for each layer. CLM calculates snow density as a function of the relative proportions of ice (mass of ice = $m_i$) and liquid water (mass of liquid water = $m_{lw}$), weighted by the snow cover fraction ($F_{sno}$) for each grid cell (Lawrence et al., 2018):

$$\rho = \frac{m_i + m_{lw}}{F_{sno} \times h_{sl}} \tag{1}$$

In practice, due to the adjusted snow cover fraction and as liquid water in the snowpack is zero until the start of melt out, the computed snow layer density simplifies to the mass of ice ($m_i$) divided by the height of the snow layer ($h_{sl}$). Changes implemented in CLM5 also include a new snow densification scheme, whereby fresh snow density is parameterised as a function of temperature and windspeed. The density of fresh snow can increase through the process of wind-driven compaction

if wind speeds exceed 0.1 m$^{-1}$ (Van Kampenhout et al., 2017). Over time, the density of the snowpack evolves as a result of the compaction processes outlined above. CLM does not allow for temperature-gradient metamorphism, and thus does not
represent the development of depth hoar layers (Van Kampenhout et al., 2017).

The computed snow layer densities are then used to calculate snow layer effective thermal conductivities (K$_{eff}$), as per Jordan (1991):

$$K_{eff} = K_{air} + \left(\left(\left(7.75 \times 10^{-5} \times \rho\right) + \left(1.105 \times 10^{-6} \times \rho^2\right)\right)\left(K_{ice} - K_{air}\right)\right) \qquad (2)$$


Values for K$_{ice}$ and K$_{air}$, the thermal conductivities of ice and interstadial air, are given in Lawrence et al. (2018). Snow (and soil) temperatures are defined for the midpoint of each layer at an hourly resolution, with the soil column consisting of 25 layers of increasing thickness (down to a depth of 49 m). Despite the simplicity of the snowpack scheme included in CLM, previous evaluation of snow heat transfer in CLM4.0 (Slater et al., 2017) suggests this modelling framework should perform
well.

## 3 Results

### 3.1 Observed meteorological, soil moisture and thermal conditions

Mean annual air temperature for 2017 - 2019 was -7.4 °C, with minimum air temperatures of -33.9 °C (2018) and -36.9 °C (2019) reached in early January (Fig 1a). The cold period was twice as long as the growing season, with consistent subfreezing
air temperatures from 10 October 2017 to 30 May 2018 (232 days) and from 23 September 2018 to 11 May 2019 (230 days). Figure 1c shows snowpack initiation in 2018 was 26 days earlier than in the previous year, with snow-on dates of 25 September 2018 and 21 October 2017 respectively. A maximum snow depth of 51 cm (2017 - 18) and 59 cm (2018 - 19) was measured at the AWS on 14 April 2018 and 11 May 2019 respectively. Snow depth from spatially distributed magnaprobe measurement showed a greater difference between the two years than at the AWS, with mean March snow depths 11 cm higher in 2018 - 19
than 2017 - 18. Magnaprobe measurements also show a higher mean March snow depth than the AWS, with March 2018 snow depths more heavily skewed than snow depths in 2019 (Fig. 2a). Snow-off date, as measured at the AWS snow depth sounder, was one week later in 2017 - 18 (30 May) than in the following year (23 May).

Soil freeze-up began with the onset of snowfall (Fig. 1b and d); 5 cm soil temperatures dropped to 0 °C on 13 October in 2017 and a month earlier on 15 September in 2018. Soil temperatures remained around 0 °C as the soil froze and released latent
heat. Soil saturation increased with depth causing a slower soil freeze-up at 20 cm than 5 cm depth in both years. A longer freeze-up in 2018 was evident from the more gradual liquid soil moisture decrease, particularly at depth (20 cm). Deeper soil (20 cm) stayed at 0 °C for longer than soil nearer the surface (5 cm), and generally remained warmer until the start of the thaw period. Minimum 2017 – 18 soil temperatures at both 5 cm (-10.9 °C) and 20 cm (-10.1 °C) depths in winter were colder than the following year (-9.5 °C and -8.2 °C), as the combined effect of earlier snowpack initiation and a deeper snow cover
prevented colder soil temperatures being reached. Variations in soil temperature in response to diurnal and synoptic weather patterns of energy inputs from the atmosphere became increasingly muted with depth in the soil column once the snowpack was established. Anomalously warm mid-winter air temperatures that approached 0 °C (22 December 2017 and 9 February 2019) or exceeded 0 °C (18 and 31 March 2019, with a rain-on-snow event occurring on the latter of these dates) had only a muted influence on the soil temperature profile (Fig. 1d), with temperatures fairly stable until sharply increasing with thaw in
early May. Soil temperatures at 5 cm increased above 0 °C for the first time on the final day of the snowmelt period in both years (Fig. 1d), with a five (2017 - 18) to seven (2018 - 19) day lag in the 20 cm soil temperatures.

## 3.2 Measured snow properties

Median density profiles from the SMP fall within the interquartile range of measured densities from volumetric sampling in
snowpits (Table 2). Snowpacks in all three campaigns (Fig. 2b - d) had a very thin surface snow layer (composed of recent
snowfall), with low near-surface snow densities (< 300 kg m$^{-3}$) rapidly increasing in the top 5 % of the snowpack. A higher
density (~ 320 kg m$^3$) wind slab layer was evident between 5 - 30 % of normalised depth from the snow surface. The next ~
10 % of the profile was a transitional section where density decreased by about 100 kg m$^3$. The lowest ~ 60 % of the profiles
is dominated by a lower density (~ 230 kg m$^3$) depth hoar layer, the density of which increases slightly towards the base of the
snowpack. Differences between median layer densities exceed the ~ 10 % sampling error associated with the use of density
cutters (Proksch et al., 2016; Conger and Mcclung, 2009), and in all but one instance, there was no overlap in the interquartile
ranges of different snow layers within a campaign (Fig. 3). Densities between 40 – 80 % of normalised depth (low density
depth hoar) are likely overestimated due to microstructural assumptions made by the algorithm of Proksch et al. (2015), which
prevent the calculation of SMP densities below 200 kg m$^3$ (see Appendix A).

The transitional section, or indurated hoar layer, with transitioning properties between wind slab and depth hoar, evident at
between ~ 30 – 40 % depth, is often difficult to capture through traditional snowpit density profiles due to the 3 cm vertical
resolution of density cutters and the layer being more defined by its crystal shape than density alone. The SMP enabled the
detection of such features due to the increased vertical resolution and vastly reduced sampling times compared to traditional
snow pits. Indurated hoar in SMP profiles was more pronounced in the 2019 campaigns; well-defined layers were not as clearly
visible in the SMP measurements from March 2018 (Fig. 2b), despite different layer densities being statistically separate in
the snowpit measurements, regardless of which year or when in the winter season the measurements were taken (Fig. 3). Ice
lenses were present in March 2018, but not during the 2019 campaigns. Throughout the course of the 2018 - 19 winter, slight
increases in the density of wind slab and depth hoar layers occurred as the snowpack developed. Late season snow densities
in both 2018 and 2019 were similar, with the exception of surface snow. The density of this layer became more variable as
each winter progressed due to the competing processes of wind compaction (increasing density) and temperature-gradient
metamorphism (decreasing density). The timing of sampling relative to fresh snowfall events, noted during both March
campaigns, also influenced measured surface snow densities.

SMP density profiles were used to parameterise profiles of thermal conductivity for the full depth of the snowpack. Patterns
in parametrized thermal conductivity profiles (Fig. 4) resemble those in SMP densities from which they were derived (Fig. 2b-
d). Surface snow thermal conductivities were low ($K_{eff-Sturm} \approx 0.1$ Wm$^{-1}$ K$^{-1}$, $K_{effs-Calonne, Jordan, Fourteau} \approx 0.2$ Wm$^{-1}$ K$^{-1}$), but sharply
increased with depth for the upper 5 % of the snowpack (Fig. 4b and c). Below this, at normalised depths of ~ 5 — 30 %,
thermal conductivity reached maximum values ($K_{eff-Sturm} \approx 0.15$ Wm$^{-1}$ K$^{-1}$, $K_{eff-Fourteau} \approx 0.25$ Wm$^{-1}$ K$^{-1}$, $K_{eff-Calonne} \approx 0.3$ Wm$^{-1}$
K$^{-1}$, $K_{eff-Jordan} \approx 0.35$ Wm$^{-1}$ K$^{-1}$). Between ~ 25 - 40 % normalised depth, thermal conductivity declined before stabilising at
minimum values ($K_{eff-Sturm} \approx 0.1$ Wm$^{-1}$ K$^{-1}$, $K_{eff-Fourteau} \approx 0.15$ Wm$^{-1}$ K$^{-1}$, $K_{effs-Calonne, Jordan} \approx 0.2$ Wm$^{-1}$ K$^{-1}$) in the lower ~ 60 %
of the snowpack. All 3 parameterisations showed similar variation in thermal conductivity with depth. Analysis of variance
showed the mean $K_{eff}$ from the Sturm et al. (1997) and Fourteau et al. (2021b) parameterisations to statistically significantly
differ from those using the parameterisation of either Calonne et al. (2011) or Jordan (1991) and each other in all three months
($F_{March2018} = 3168$, $F_{Jan2019} = 656$, $F_{March2019} = 636$). No significant difference was found between the Calonne et al. (2011) or
Jordan (1991) parameterisations in either of the 2019 campaigns. All statistical tests herein gave a p-value less than 0.001,
denoting significance at the 99.9% level.

Profiles of snowpack thermal conductivity were temporally consistent, with similar shape and values in January and March
2019. In March 2018, the amplitude of the thermal conductivity profiles was less pronounced than January and March 2019,
particularly for the parameterisation of Sturm et al. (1997). We recognise that the thermal conductivity of a snowpack is
dependent on more than just its density (Sturm et al., 2002), with other factors such as snow microstructure and temperature

also having an influence i.e. (Calonne et al., 2011) but these profiles still provide novel insights and a useful first-order approximation of snow heat transfer for model evaluation.

### 3.3 Modelled snowpack properties and comparison with observations

Simulated snow depths (Fig. 5a and d) were consistently lower than observations (from either Magnaprobe measurements (mean value) or the acoustic sounder depth on the 31 March at the AWS; Fig. 1b, Table 1). Timing of simulated snowpack
accumulation leads to an effective snow depth in 2018 - 19 ($S_{depth,eff}CLM_{2018-19}$ = 66 cm) more than double that in 2017 - 18 ($S_{depth,eff}CLM_{2017-18}$ = 24 cm) with earlier snow onset allowing a greater degree of soil insulation. Simulated snow onset (11 October) and melt-out dates (25 May) were both approximately a week earlier than observed at the AWS in 2017 - 18; for the following year the length of this offset was reduced to just one day. Observations of effective snow depth ($S_{depth,eff}Obs_{2017-18}$ = 57 cm, $S_{depth,eff}Obs_{2018-19}$ = 101 cm) similarly reflect greater insulation of the soil surface in 2018-19 compared to 2017-18.

The physical properties of the simulated snow layers do not correspond to observations, with the number and thickness of snow layers only a function of overall snowpack depth. Figs. 5 b & e show three (or four) relatively homogenous layers, with a slight increase in density with depth. The highest mean (329 kg m$^3$) and median (340 kg m$^3$; Table 2) density are found in third snow layer (dark blue in Fig. 5).

This is in contrast to the three observed layers (surface snow, wind slab and depth hoar) consistently identified in the snowpit
observations. Similar to other snow models (Domine et al., 2016; 2019) the physical characteristics of the depth hoar layer at the base of the snowpack (large faceted grains; low density) are not clearly distinct from an overlying wind slab layer (small rounded grains; high density). This is the result of the lack of representation of depth hoar layer development in CLM (Van Kampenhout et al., 2017). These discrepancies between modelled and measured snow density and stratigraphy negatively impact the simulation of $K_{eff}$, as layer thermal conductivities were dependent on density of each layer (Eq. 2).

CLM overestimated the thermal conductivity of tundra snowpacks compared to in-situ measurements using needle probes or estimated from SMP profiles (Fig. 6a). Median simulated snow thermal conductivities (0.34 Wm$^{-1}$ K$^{-1}$) were at least three times greater than either needle probe measurements (0.08 Wm$^{-1}$ K$^{-1}$) or SMP-derived estimates using the Sturm parameterisation($x_{Keff-Sturm}$ = 0.11 Wm$^{-1}$ K$^{-1}$), with the median thermal conductivity using the Calonne, Fourteau and Jordan approximations still lower ($x_{Keff-Calonne}$ = 0.25 Wm$^{-1}$ K$^{-1}$, $x_{Keff-Fourteau}$ = 0.21 Wm$^{-1}$ K-1, $x_{Keff-Jordan}$ = 0.27 Wm$^{-1}$ K$^{-1}$) than simulated
thermal conductivities. SMP $K_{eff}$ parameterisation from Sturm et al. (1997; derived from snow measurements in the Alaskan arctic), are closer to values from needle probe measurements than SMP $K_{eff}$ derived using Calonne et al. (2011) (Fig. 6a). The modelled thermal conductivity of simulated snow layers was relatively homogenous between layers in contrast to thermal conductivities derived from either the SMP (Fig. 4) or the needle probe measurements (Table 2). Analysis of variance only shows simulated snow layer thermal conductivities significantly differ from that of the surface layer (F = 39.74). Needle probe
measurements of the depth hoar layer had low thermal conductivities (0.05 Wm$^{-1}$ K$^{-1}$), with a slight increase in mean thermal conductivity for indurated hoar (0.09 Wm$^{-3}$ K$^{-1}$) and a further increase for the mean wind slab thermal conductivity (0.20 Wm$^{-1}$ K$^{-1}$). Distributions of simulated snow thermal conductivities were statistically significantly different from all measurement methods at the 0.01 level using a Kruskal-Wallis test. Differences between the distribution of needle probe measurements and SMP with the Sturm parameterisation were not statistically significant.

**3.4 Improving simulated soil temperatures, snow thermal conductivity and snow heat transfer**

Simulated soil temperatures were considerably colder than observations (RMSE = 5.0 °C, Bias = - 2.2 °C), especially during the maximum annual duration of continuous simulated snow cover (15 Sept – 31 May; RMSE = 5.8 °C). Two approaches were taken to reduce simulated snow thermal conductivities, both of which resulted in warmer soil temperatures closer to observed values (Fig. 7a & b).

In order to see how results from the SMP (Fig 6a) manifested in simulations of soil temperature from CLM, we re-ran the model substituting the default parameterisation for snow thermal conductivity (Eq. 1; Jordan (1991)) for those of Sturm et al. (1997), Calonne et al. (2011) and Fourteau et al. (2021b). The Sturm parameterisation resulted in lower simulated thermal conductivities (Fig. 6b) and closer temperatures to observations (Fig. 7b; RMSE = 2.5°C). Soil temperatures in 2017-18 were still too cold regardless of parameterisation used, likely due to model underestimation of snow depth (Fig 7c). As for the SMP, thermal conductivity values derived using the Calonne and Fourteau parameterisations are closer to the default Jordan (1991) parameterisation than those derived using the Sturm et al. (1997) parameterisation (Fig 6b). The impact of either of these parameterisations on simulated wintertime soil temperatures is limited (Fig 7a), particularly that of Calonne et al. (2011) which reduces the RMSE by only 0.2 °C. However, all 3 alternative parameterisations tested do show an improvement in simulated snow thermal conductivities (Fig 6b) and soil temperatures (Fig 7a), with an increase in the value of the SHTM in each case. We also tested the application of a multiplier ($\alpha$) to the ice content term in Eq. 1:

$$\rho = \frac{(\alpha \times m_i) \times m_{lw}}{F_{sno} \times h_{sl}} \tag{3}$$

Although appearing to be a function of density, this multiplier is added separately from the calculation of layer snow densities, and only feeds into the calculation of snow thermal conductivity, and thus snow mass is conserved. Values of $\alpha$ were chosen which would reduce simulated densities to the range of observed values, with an $\alpha$ of 0.65 giving the $K_{eff}$ for snow with a density between the interquartile range of observed values for all snow types (73 – 365 kg m$^3$). A set of sensitivity tests were then carried out where the value of $\alpha$ was iteratively changed from 0.75 to 0.25 in 0.05 increments . As the RMSE and the SHTM quantify changes over slightly different time periods (RMSE = entire winter, SHTM = Oct – March), different metrics may imply different adjustments give the best model performance. In 2018-19, a value of $\alpha$ between 0.65 and 0.6 resulted in the optimal model performance , with a SHTM value of 0.991 (or 0.979) and a RMSE of 1.5 °C (or 1.2 °C). However, a smaller value of $\alpha$ was required for best model performance in 2017-18, with an $\alpha$ of 0.4 giving the lowest RMSE of 1.6°C and highest SHTM of 0.986. Reducing simulated snow density in Eq. 3 ($0.3 \geq \alpha \geq 0.55$) below the lowest quartile of observed values was required to increase soil temperatures to the observed range, particularly for 2017 – 18 where wintertime minimum soil temperatures are up to 12.8 °C warmer relative to the baseline model run (Fig. 7b). Different $\alpha$ will better fit different years of the simulation, though using the same best-fit value of $\alpha$ for the entire model run can still give good model performance, with a maximum value for the SHTM of 0.987 for an $\alpha$ of 0.40

Errors in the timing and depth of simulated snow cover (Fig. 7c) impact the magnitude of insulation it provides, and thus the best-fit value of $\alpha$ (Fig. 8). A multiple linear regression was undertaken to quantify the influence of snow depth and snow depth error on the value of the best fit correction factor, for the period from snow onset to the start of simulated snow melt (when the simulated snow cover fraction was equal to one). This showed errors in the simulated snow depth can be compensated by a greater adjustment to snow thermal conductivity (Fig 8b):

$$\alpha = 0.22 + 1.14S - 0.26E + 0.55SE \tag{4}$$

where $S$ equals the simulated snow depth, and $E$ equals the simulated snow depth error. Best fit correction values were strongly related to snow depth ($R^2$ = 0.77, RMSE = 0.066), with different values of $\alpha$ more appropriate for deep (> 25 cm, $\alpha \approx 0.6$) and shallow (< 15 cm, $\alpha \approx 0.3$) snow (Fig. 8a).

## 4 Discussion

### 4.1 Variability of snow thermal properties

SMP profiles, processed as detailed in Appendix A, produced snow layer densities closely matched to density cutter measurements at TVC (Fig. 3) and consistent with measurements from other Arctic and sub-Arctic environments, e.g. $\rho_{SS}$ = ~ 100 kg m$^3$, $\rho_{WS}$ = 300 - 500 kg m$^3$, $\rho_{DH}$ = 150 - 250 kg m$^3$ in Barrere et al. (2017); Benson and Sturm (1993); Derksen et al. (2014); Domine et al. (2002; 2012; 2016). SMP profiling has considerably increased the vertical resolution of density measurements and vastly reduced sampling times compared to traditional snowpits, enabling a far greater number of measurement profiles to be made across a wider distribution of snowpack conditions. Deriving profiles of thermal conductivity for the full depth of the snowpack, as facilitated by the SMP, is a novel approach, with most previous studies of snow thermal conductivity based on values sampled at a resolution of ~ 5 - 10 cm (Domine et al., 2012; 2015; 2016; Gouttevin et al., 2018; Morin et al., 2010).

Depth normalisation of SMP profiles (n > 200 per measurement campaign) allowed comparison of snow properties with varying absolute depth. Snow depth distributions from all campaigns matched the shape and median values of tundra snow depths acquired across a ~ 1500 km traverse as described in Derksen et al. (2009), which suggests transferability across wider Arctic tundra regions. Relative depth profiles of density at TVC remain consistent for all sampling campaigns, regardless of overall snowpack depth. Densities in the portion of the depth hoar layer located between 40 - 80 % depth were likely overestimated (although SMP estimates remain within the interquartile range of snowpit measurements) due to an assumption of heteroscedasticity made by the algorithm of Proksch et al. (2015), which may not apply for a material as anisotropic as depth hoar (Fig. A2). Additionally, pressure exerted on the ice matrix by the SMP may have caused wider collapse of the weak depth hoar structure during measurement (although SMP operators are easily able to profiles that are obviously affected by depth hoar collapse). As a result, the force required to penetrate the snow may be reduced (potentially below the detection limit of the SMP) in the gaps where the ice matrix has collapsed; required penetration force will conversely increase towards the base of the snowpack where the collapsed depth hoar has accumulated. This, plus an increased probability of SMP-vegetation interactions at the base of the snowpack, is likely the cause of density (and density-derived $K_{eff}$) increases in the lower ~ 20 % of all profiles. While exact impact of ice matrix collapse in depth hoar is not possible to quantify directly, this limitation is not without comparison in other direct, contact measurements of snow properties such as volumetric sampling of density (Conger and Mcclung, 2009; Proksch et al., 2016) and μ-CT (Zermatten et al., 2011).

The higher vertical resolution of SMP density profiles (1.25 mm, or 0.25 % of snowpack depth) relative to traditional snowpit measurements (3 cm) allows snowpack features to be much more finely resolved (Calonne et al., 2020; King et al., 2020b; Proksch et al., 2015). Moving away from bulk sampling of layers with boundaries defined by abrupt binary transitions as identified by traditional stratigraphic techniques, to more continuous profiles enables features such as indurated hoar, typically a subtle transitional layer, to be captured and quantified (Pielmeier and Schneebeli, 2003; Proksch et al., 2016). Higher resolution measurements (μ-CT, SMP) of continuous profiles are increasingly implemented (e.g., Proksch et al. (2016); Calonne et al. (2020); Wagner et al. (2021)) but this conceptualisation of snow as a continuous profile rather than a series of discrete layers is not yet implemented in snowpack modelling, excepting the test case outlined by Simson et al. (2021).

### 4.2 Evaluation of snowpack and soil temperature simulations

Density profiles of Arctic snow from physical snow model simulations are inverted relative to observations, exhibiting low density snow in the upper part of the snowpack and high density snow at the base, similar to what would be expected in alpine environments (Barrere et al., 2017; Domine et al., 2019). CLM is no exception, with the model producing three to four layers of uniformly high density snow, rather than a low density snow layer adjacent to the ground overlain by a higher density slab layer. Consequently, simulated density profiles are not representative of field measurements and the overall bulk density of

the snowpack is overestimated. This is common of other snow models of similar physical complexity, e.g. ISBA-ES (Barrere et al., 2017), and higher complexity, e.g. SNOWPACK (Bartelt and Lehning, 2001) and Crocus (Vionnet et al., 2012), because they do not account for unique arctic processes (Domine et al., 2016; 2019), such as the snowpack vapour flux necessary to form depth hoar. As $K_{eff}$ is simulated as a function of density, when models are unable to accurately describe the density profiles of Arctic snowpacks, this has a negative impact on how well $K_{eff}$ can be simulated (Gouttevin et al., 2018). $K_{eff}$ values from CLM are not only overestimated relative to field measurements, but also in comparison to simulations from more complex snow models in similar environments (Barrere et al., 2017; Domine et al., 2019). These problems with thermal conductivity simulations subsequently impact soil temperatures, with similar issues found for simulations of Arctic snowpacks using other models, i.e., Crocus, SNOWPACK, ISBA-ES (Barrere et al., 2017; Domine et al., 2016; 2019; Royer et al., 2021b).

The impact of snow insulation on soil temperatures is dependent on both the depth and thermal conductivity of the snowpack (Gouttevin et al., 2012), as well as the timing of snow accumulation (Lafrenière et al., 2013). The start of the snow season is particularly important because erroneous modelled heat exchanges between air, snow and soil influence soil and snowpack properties and development, which are carried forward until the end of the snow season (Sandells et al., 2012). Temperature differences between soil and air induce a strong snowpack temperature gradient, leading to depth hoar formation and thus determining the structure of the snowpack and its capacity to insulate the soil (Domine et al., 2018).

### 4.3 Impact of approaches to correct snow thermal conductivity

Prescribing simulated snow thermal conductivity to a more physically representative value leads to an improvement in simulating soil temperatures in tundra environments, compared to both the findings herein and the permafrost model used in Yi et al. (2020). Cook et al. (2007) also found that reducing simulated snow thermal conductivity to the lower end of observed values (0.1 Wm$^{-3}$) reduced soil temperature biases in an older version of CLM (CLM3.0). It has also been suggested that the simulation of wintertime soil temperatures at TVC may also be influenced by simulated soil properties and the impact of the snow cover on soil moisture content (Haagmans, 2021); bias is unlikely to be completely eliminated solely as a result of changes to snow thermal conductivity.

The impact of alternative parameterisations of snow thermal conductivity on simulated soil temperatures was tested, with a reduction in the RMSE and an improvement in the SHTM found for all 3 alternative parameterisations tested. Changing the parameterisation of snow thermal conductivity in CLM from that of Jordan (1991) to that of Sturm et al. (1997) gives the largest improvement to the simulation of both snow thermal conductivity values and underlying soil temperatures. Use of the Sturm et al. (1997) thermal conductivity parameterisation also improved soil temperature simulation in Crocus (Royer et al., 2021b), with a RMSE of 2.5 °C for soil temperatures from Crocus and CLM. The Sturm et al. (1997) parameterisation demonstrates transferability between tundra sites, having been derived from thermal conductivity measurements in the Alaskan Arctic and successfully applied to both CLM and SMP measurements at TVC. Although concern has been raised that the parameterisation of Sturm et al. (1997) may not be physically representative, we feel this provides the most feasible solution to improving soil temperature simulations in CLM given the sizeable improvement in RMSE and its use in more physically representative land surface models (Royer et al., 2021b).

Application of the correction factor $\alpha$ improves the simulation of soil temperatures, increasing the value of the SHTM by up to 0.3. The impact of differences between simulated and observed snow depth can be compensated by a greater adjustment to snow thermal conductivity (Figs. 7b & 8). This bias compensation between underestimates of snow depth and underestimates of snow thermal conductivity is also seen in other land surface models, e.g. JULES, LPJ-GUESS (Wang et al., 2016). However, as discrepancies between observed and simulated snow depth can vary considerably between years, this results in a best-fit correction factor value which also changes between years. These findings indicate that thermal conductivity correction factors are not the solution to soil temperature biases in models like CLM.

Differences between absolute and effective snow depths from both the model and the observational record highlight the importance of the early season snowpack in regulating soil temperatures for the entire snow season. Simulations are sensitive to latent heat release during soil freeze-up, which maintains soil temperatures close to 0 ℃ for an extended period of time at the beginning of the winter (Yi et al., 2019). At this time, the soil thermal regime is also more sensitive to snow depth as snow depths are lower and have not yet reached a point where their insulative capacity has become saturated (Zhang, 2005; Lawrence and Slater, 2009; Slater et al., 2017), therefore a stronger correction is needed when snow cover is below ~25 cm. Shallow snowpacks are likely to consist of a lower proportion of wind slab (Rutter et al., 2019) and thus their microstructural properties are less accurately represented by CLM, which does not simulate depth hoar (Van Kampenhout et al., 2017), stipulating the need for a larger adjustment to $\alpha$. We note that issues in simulating the initial accumulation of the snowpack are likely linked to uncertainties in the forcing data caused by measurement limitations surrounding the use of precipitation gauges in tundra environments (Smith, 2008; Watson et al., 2008; Pan et al., 2016).

Regardless of approach, these changes to the model are most applicable where snowpack structure is considerably influenced by depth hoar, as can be approximated by grid-cell plant functional type or climatology (Royer et al., 2021a; Sturm and Liston, 2021).

Ekici et al. (2015) suggests that representation of snow thermal conductivity in land surface models is less important for accurate simulation of soil temperatures than other processes not currently well represented in most land surface schemes, such as blowing snow and depth hoar formation. Further improvements in SHTM in future iterations of CLM will require a physically representative approach to snow density and thermal conductivity through explicit inclusion of vapour transport within the snowpack, currently under development in stand-alone snow microphysical models (Fourteau et al., 2021a; Jafari et al., 2020; Schürholt et al., 2021). However, this presents computational and mathematical challenges, as outlined in Jafari et al. (2020). The inclusion of physically representative parameterisations of snow properties in land surface models, such as that of Royer et al. (2021b) where the densities of lower snow layers are not allowed to exceed a maximum observation-based threshold, are more likely in the near future than the explicit representation of snowpack vapour transport. Meanwhile the substitution of the Sturm et al. (1997) thermal conductivity parameterisation provide a computationally efficient compromise, reducing both the value of $K_{eff}$ and the cold bias of simulated wintertime soil temperatures considerably (RMSE reduction of 3.3 ℃).

Model underestimates of soil temperatures follow through into calculations of soil respiration, further contributing to uncertainties surrounding estimates of wintertime carbon flux (Natali et al. 2019) and suggesting that such modelled values are likely to be an underestimation of the true magnitude of these fluxes. Being able to accurately model fluxes outside of the growing season is important as these make a considerable contribution to the annual carbon budget (Natali et al 2019; Schuur et al 2021). A low soil temperature bias due to poorly simulated snow insulation also has consequences for predicting the evolution of permafrost (Barrere et al., 2017; Burke et al., 2020) and resultant carbon emissions when it degrades (Peng et al., 2016).

## 5 Conclusions

A new recalibration to derive profiles of tundra snow density and thermal conductivity from SMP profiles of penetration force is presented, with resulting densities and thermal conductivities then used to evaluate the performance of CLM5.0. SMP-derived density profiles show good agreement with measured snow layer densities at TVC. Comparison of measured snowpack properties from in situ SMP and needle probe techniques with simulations show the model tends to overestimate snow layer thermal conductivities by a up to factor of three, with implications for how well wintertime soil temperatures are simulated. Alternative relationships between snow density and snow thermal conductivity were considered, all of which improved the simulation of wintertime soil temperatures (RMSE reduction of 0.2 – 3.3 ℃). Reducing simulated thermal conductivities

through the use of a correction factor ($\alpha$) also improves simulation of soil temperature (RMSE reduction of 3.7 °C for an $\alpha$ of 0.45). The optimal magnitude of this reduction is strongly linked to snow depth (with a greater reduction needed for shallower snowpacks). Different optimal correction factors for different snow seasons illustrate the limitations of this approach, but the results are still instructive as a diagnostic for model sensitivity to the treatment of snow thermal conductivity

Further improvements to simulated snow properties will require more explicit representation of key processes not currently accounted for in CLM, chiefly the formation of depth hoar. A more physically representative snowpack should also improve simulation of wintertime soil thermal conditions. Snowpack vapour kinetics are not currently included within global land surface models, which also have to consider a large variety of other processes and avenues for future development (Blyth et al., 2021; Fisher and Koven, 2020), although developments are being made to consider these in complex microscale snow physics models. Empirical scaling of snow thermal conductivity provides a computationally efficient interim solution with a similar impact on soil temperatures as the explicit representation of a large depth hoar fraction in point-scale simulations by Zhang et al. (1996), but the value of the required scaling factor changes with snow depth. Different parameterisations of snow thermal conductivity also improve simulation of soil temperatures, with that of Sturm et al. (1997) more appropriate for Arctic snowpacks (RMSE reduction of 3.3 °C) than that of Jordan (1991) which is used by default in CLM. Improving the accuracy with which Arctic wintertime soil temperatures can be simulated may help to reduce sizable uncertainties (Natali et al., 2019) surrounding current projections of wintertime carbon fluxes.

## 6 Appendix A: SMP Processing

Differences between study environments (the original SMP coefficients were not derived for tundra snow) and SMP hardware for different versions of the SMP used by Proksch et al. (2015) and this study required new coefficients to be derived in order to relate penetration force to snow density. Methods from King et al. (2020b) were adapted to recalibrate SMP measurements from TVC in January and March 2019, described in detail in Fig. A1. Co-incident SMP profiles and snowpit density measurements were available at 36 locations across the TVC catchment. A K-folds process is then used to derive new coefficients ($a$ – $d$; Table A1) for Eq. A1 (Eq. 9 in Proksch et al. (2015)):

$$\rho_{SMP} = \boldsymbol{a} + \boldsymbol{b}ln(\tilde{F}) + \boldsymbol{c}ln(\tilde{F})L + \boldsymbol{d}L \tag{A1}$$

where $\tilde{F}$ is the median force value over the vertical distance where density is calculated and $L$ is the element size, the distance between points where force is exerted by the SMP - approximately the distance between snow grains (Löwe and Van Herwijnen, 2012). Individual pairs of SMP derived and snowpit measured densities above the 95th percentile of absolute error were removed (Fig. A1 – Step 7), and the K-folds recalibration repeated (Step 8) to produce revised coefficients to recalibrate the entire SMP dataset (Step 9). This process was iterated until paired SMP-snowpit profiles with an $R^2$ of less than 0.7 were removed. Poor fitting between some paired SMP-snowpit profiles was due to the spatially heterogeneous nature of the snowpack (King et al., 2020b), as microtopographic variation in hummocky tundra can lead to considerable sub-metre snowpack variability. Coefficients (Table A1) were ultimately derived from 21 paired SMP-snowpit density profiles; 16 from the January 2019 campaign, and 5 from the March 2019 campaign ($R^2 = 0.88$, p < 0.001). These coefficients give a RMSE of 25.2, compared to an RMSE of 125 for those of Proksch et al. (2015).

These coefficients were used to calculate density profiles for all 640 profiles from the 2019 campaigns (Fig. A1 – Step 12). Densities for SMP profiles from the March 2018 campaign were also derived from these, but measurements from this campaign were not included in the recalibration dataset. Metrics were calculated over a 2.5mm sliding window with 50 % overlap, (ie. 1.25mm resolution) as per Proksch et al. (2015). Profiles of thermal conductivity were then calculated from SMP densities, using the density: $K_{eff}$ relationships derived by Sturm et al. (1997), Calonne et al. (2011), Jordan (1991), and Fourteau et al.

(2021b) (Fig. A1 - Step 17). It is important to note that the thermal conductivity of a snowpack is dependent on more than just its density (Sturm et al., 2002; Fourteau et al., 2021b), but these parameterisations provide a useful first-order approximation. Prior to recalibration, negative force values were removed from the SMP profiles. These are erroneous values which can occur in the SMP output when ice gets caught in the cog wheel of the SMP or if part of the instrument is damaged (Lutz, 2009). Buried vegetation may also be present in the lower part of tundra snowpacks, and interaction between SMP and dense shrubs or branches may cause the SMP signal to overload and affect the quality of lower sections of the profile. A normalised percentage depth scale (with profiles rescaled to a resolution of 0.25 % of total depth using linear interpolation) was used to compare SMP-derived profiles of density and $K_{eff}$ from different snow depths (Steps 15 and 18). Any negative densities or thermal conductivities were removed during the depth normalisation process.

Recalibrated density profiles from the SMP do not produce values below 200 kg m$^3$, despite observations of lower snow densities in Arctic depth hoar, including some from this campaign (Fig. 2). Figure A2b shows a large spread in the value of $L$ for the depth hoar samples, over a relatively small set of snowpit densities. Large element sizes, or distances between snow grain failures, are not unexpected in depth hoar but this results in a low signal to noise ratio (King et al., 2020b). Figure A2 shows the relationship between $\tilde{F}$ and $L$ is not heteroscedastic as initially assumed, leading to an overestimation of the density (and density-derived $K_{eff}$) of this layer. Proksch et al. (2015) state that their model does not yet fully account for the anisotropic structure of some snow types, which is of particular relevance to depth hoar.

## 7 Appendix B: Model Spin-up

In order to determine the amount of model spin-up required for soil temperatures to equilibrate, iterative runs of PTCLM with an additional year of spin-up were undertaken from 1 January 2017 to 1 January 2013. Soil temperatures throughout the soil column were compared; 3 depths are shown in Fig. B1. Internal system variability results in a difference of ~ 1 °C between model runs, with a minimum of 2 years of spin-up required for $K_{eff}$ adjusted runs to converge at a 10cm soil depth. Deviation between different spin-up start times takes longer to level out deeper in the soil column, but as we only examine soil properties within the top 20cm of the soil column, we feel this length of spin-up is sufficient. Changes to snow thermal conductivity were evident at all depths in the soil profile, and have an impact on the thickness of the active layer with seasonal thawing seen to a depth of 1.7 m (Fig. B1b), in comparison to 1.35 m for the unadjusted CLM runs and the 1 m active layer depth reported by Grünberg et al. (2020).

## Code & Data availability

Code and data to produce figures is available at: https://github.com/V-Dutch/TVCSnowCLM

## Author Contribution

Investigation, Formal Analysis, Writing - Original Draft preparation; VRD. Supervision; NR, LW, MS, CD, RK. Data acquisition; NR, CD, RE, JK (TVC Snow Data); BW, GHG, OS, JB (TVC Meteorological Observations). Data Planning; PM. Software; JK, MS (SMP); LW (CLM). Funding acquisition; NR. All authors were involved in reviewing and editing.

## Competing Interests

Some authors are members of the editorial board of The Cryosphere. The peer-review process was guided by an independent editor, and the authors have also no other competing interests to declare.

**Acknowledgements**

530    VRD was funded by an RDF Studentship from Northumbria University and the Northern Water Futures project. NR and LW were supported by NERC Grant NE/W003686/1. NR and RE were supported by NERC Arctic Office United Kingdom & Canada Arctic Partnership Travel Bursaries. Funding for JB was provided from the Helmholtz Association in the framework of MOSES (Modular Observation Solutions for Earth Systems). This project was conducted with approval issued by Aurora Research Institute – Aurora College (License Nos. 16237 & 16501). The authors would like to acknowledge that this study

535    occurred within the Inuvialuit Settlement Region located in the Western Canadian Arctic.

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

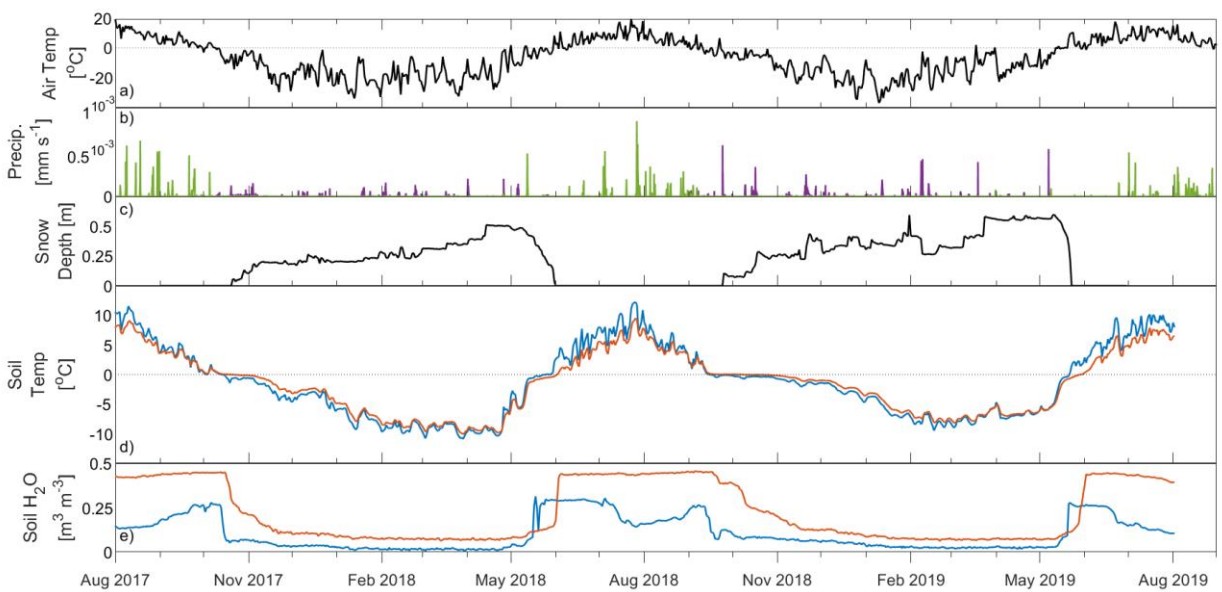

**Figure 1: Daily averaged meteorological and soil conditions at Trail Valley Creek from 1 August 2017 to 31 August 2019; a) 2 m air temperature, b) precipitation: snow (purple) and rain (green), c) snow depth, d) soil temperatures at depth of 5 cm (blue) and 20 cm (orange) and e) volumetric soil water content at 5 cm (blue) and 20 cm (orange) depths.**


| | Snow Depth [cm] | |
| --- | --- | --- |
| | March 2018 | March 2019 |
| AWS | 35 | 56 |
| Magnaprobe | 33 ± 15.7 (n = 14,966) | 44 ± 14.4 (n = 8541) |
| CLM | 18 | 34 |

**Table 1: End of March snow depth summary. Mean and standard deviation of spatially distributed measurements with a sample size greater than n = 1 are shown, otherwise the daily value for 31st March is shown.**


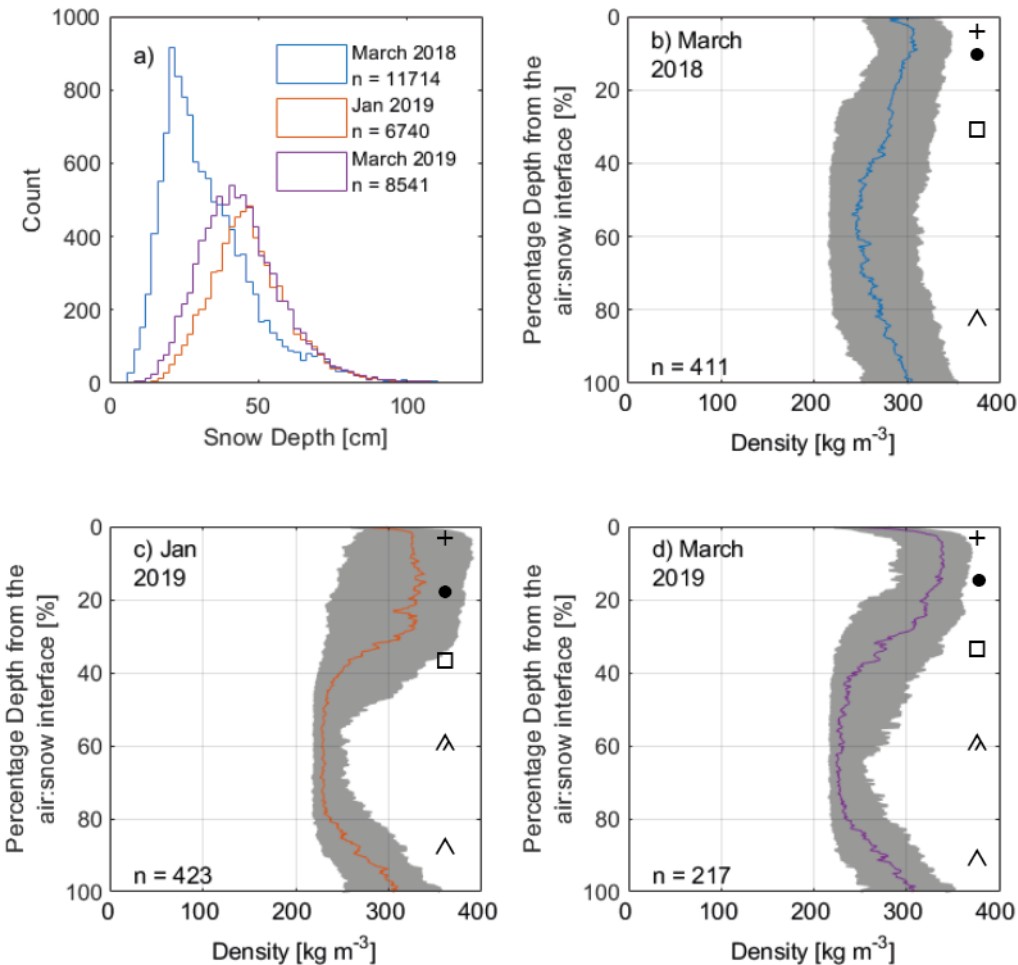

**Figure 2: a) Frequency distributions of magnaprobe depths for each sampling campaign where Snow MicroPenetrometer (SMP) measurements are available; b-d) Profiles of median SMP-derived densities (colour-coded for the respective campaigns (Fig. 3a); interquartile range shaded in grey), with snow stratigraphy as per Fierz et al. (2009) superimposed.**

| | | # of Layers | Layer | Median | Interquartile Range | |
|---|---|---|---|---|---|---|
| Density [kg m⁻³] | Snowpit Obs. | 3 | Surface Snow | 89 | 73 | 152 |
| | | | Wind Slab | 334 | 300 | 365 |
| | | | Depth Hoar | 249 | 228 | 270 |
| | CLM | 4 | 1 | 270 | 209 | 328 |
| | | | 2 | 328 | 285 | 346 |
| | | | 3 | 340 | 282 | 346 |
| | | | 4 | 309 | 291 | 326 |
| Thermal Conductivity [W m⁻¹ K⁻¹] | Snowpit Obs. | 3 | Surface Snow | - | - | - |
| | | | Wind Slab | 0.20 | 0.15 | 0.28 |
| | | | Depth Hoar | 0.05 | 0.04 | 0.06 |
| | CLM | 4 | 1 | 0.25 | 0.17 | 0.35 |
| | | | 2 | 0.35 | 0.28 | 0.38 |
| | | | 3 | 0.37 | 0.27 | 0.38 |
| | | | 4 | 0.32 | 0.29 | 0.35 |

**Table 2: Summary of modelled (CLM) and measured snow densities and thermal conductivities (from the manual density profiles and the needleprobe, respectively).**

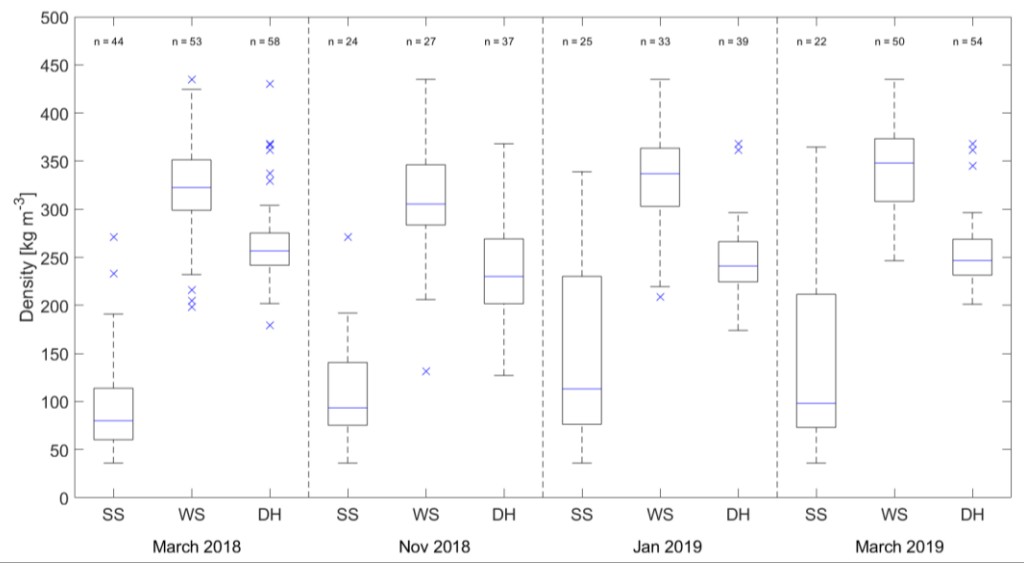


**Figure 3: Distributions of measured layer densities (SS = surface snow, WS = wind slab, DH = depth hoar) from four sampling campaigns: box (interquartile range), blue line (median), and whiskers (dashed lines) extend from the end of each box to 1.5 times the interquartile range; outliers beyond this range (blue crosses).**

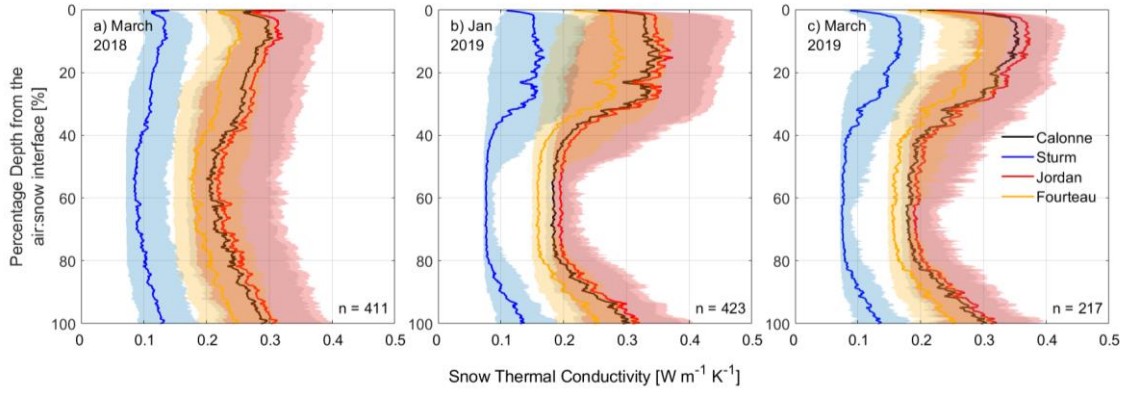


**Figure 4: Median thermal conductivity profiles (lines) and interquartile range (shaded areas) approximated from SMP densities, using the parameterisations of Calonne et al. (2011) in black/grey, Sturm et al. (1997) in blue, Jordan (1991) in red and Fourteau et al. (2021b) in yellow.**


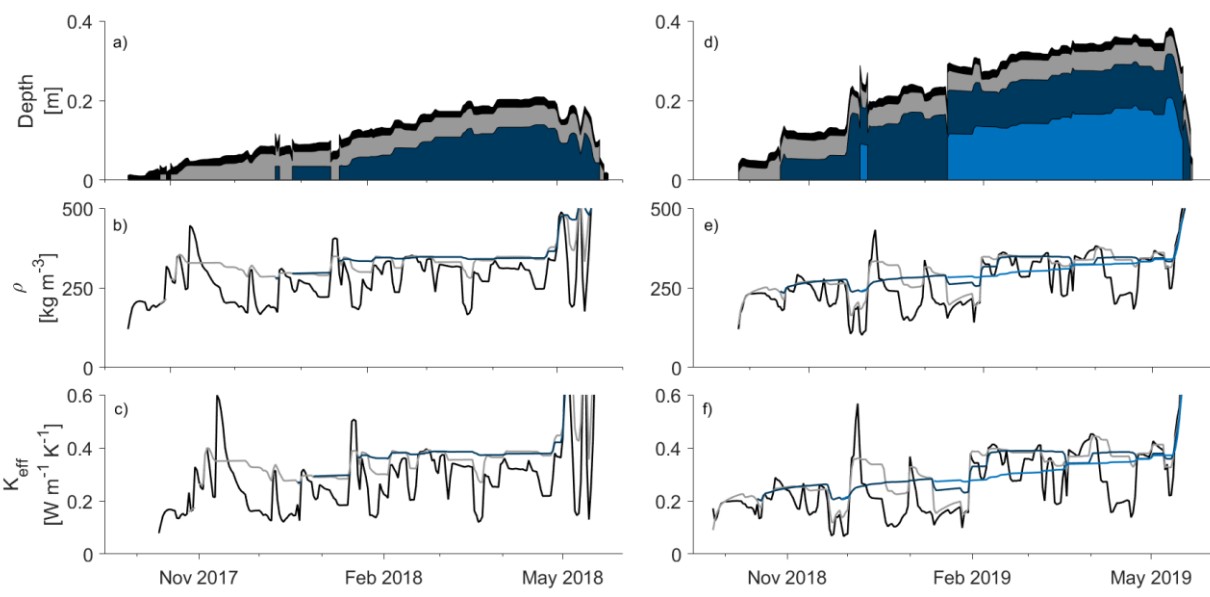

**Figure 5: Simulated snow layers and their properties for winter 2017-18; a) simulated snow layer thicknesses, b) snow layer densities, c) snow layer thermal conductivities; d) to f) as before but for winter 2018-19. Each colour represents**
**a different computational snow layer.**

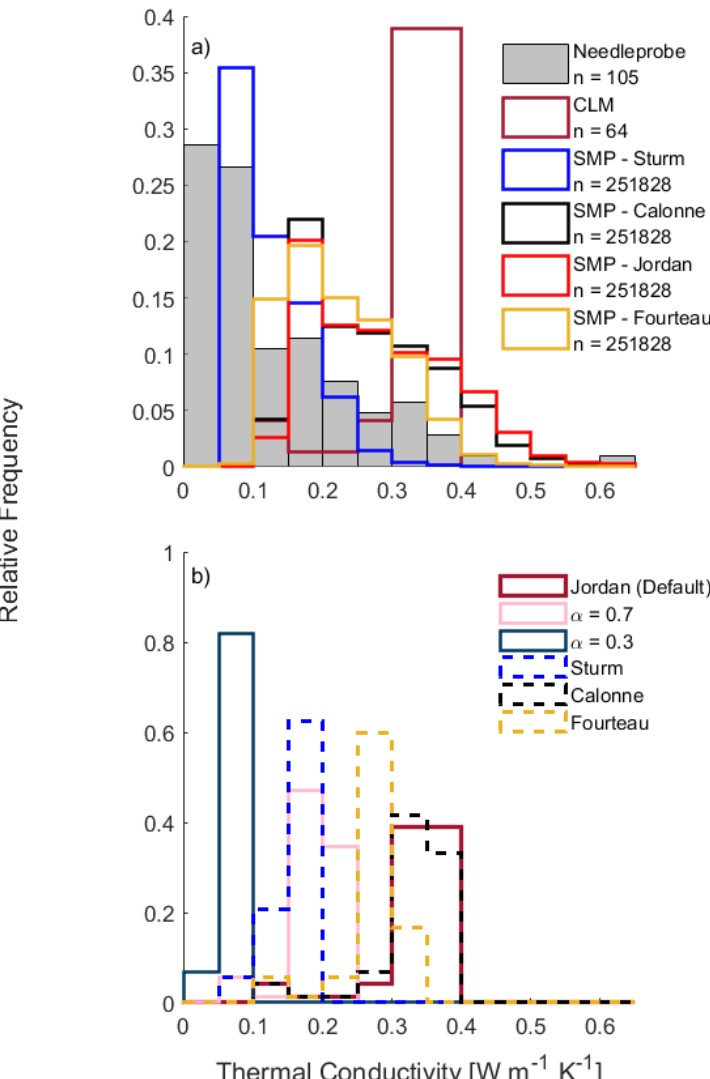

**Figure 6: a) Histograms of measured and simulated thermal conductivity from March 2018 and 2019 sampling campaigns; dashed lines show the four different thermal conductivity parameterisations applied to the SMP densities, 825 b) Sensitivity testing of simulated thermal conductivities for the same time period using both the default CLM snow thermal conductivity parameterisation, application of the α correction (solid lines), and alternative snow thermal conductivity parameterisations ( (Calonne et al., 2011; Fourteau et al., 2021b; Sturm et al., 1997) ; dashed lines). Note the different scales on the y-axes.**

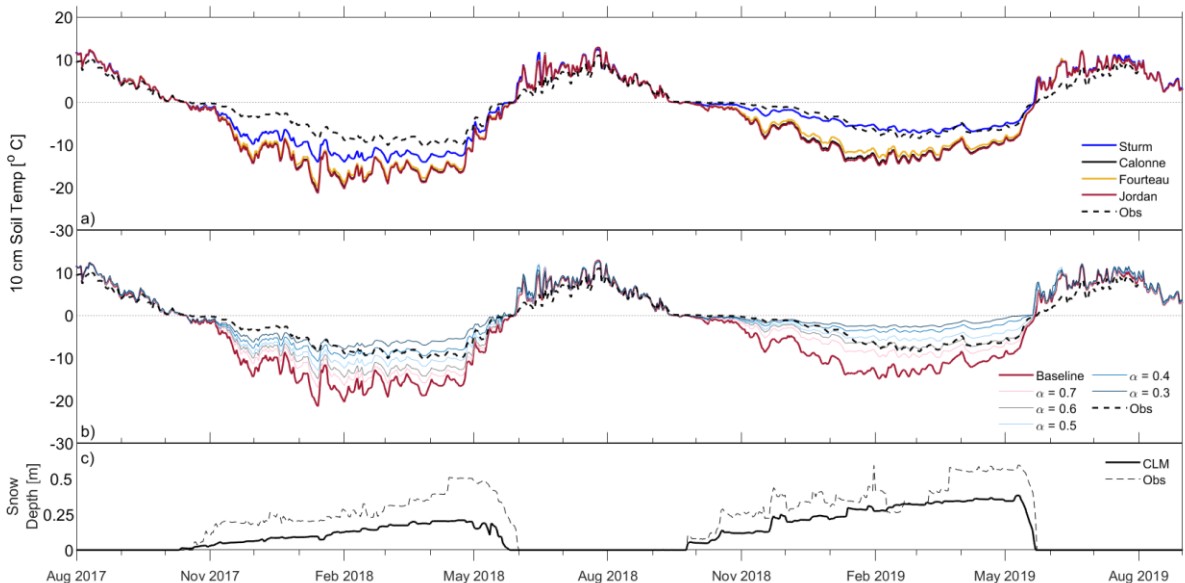

**Figure 7: a) Simulated 10cm soil temperature timeseries using the 4 different parameterisations of snow thermal conductivity compared to field measurements, b) Timeseries of 10cm soil temperatures when using different values of the correction factor $\alpha$ compared to field measurements, c) Observed and simulated snow depths for the same time period.**


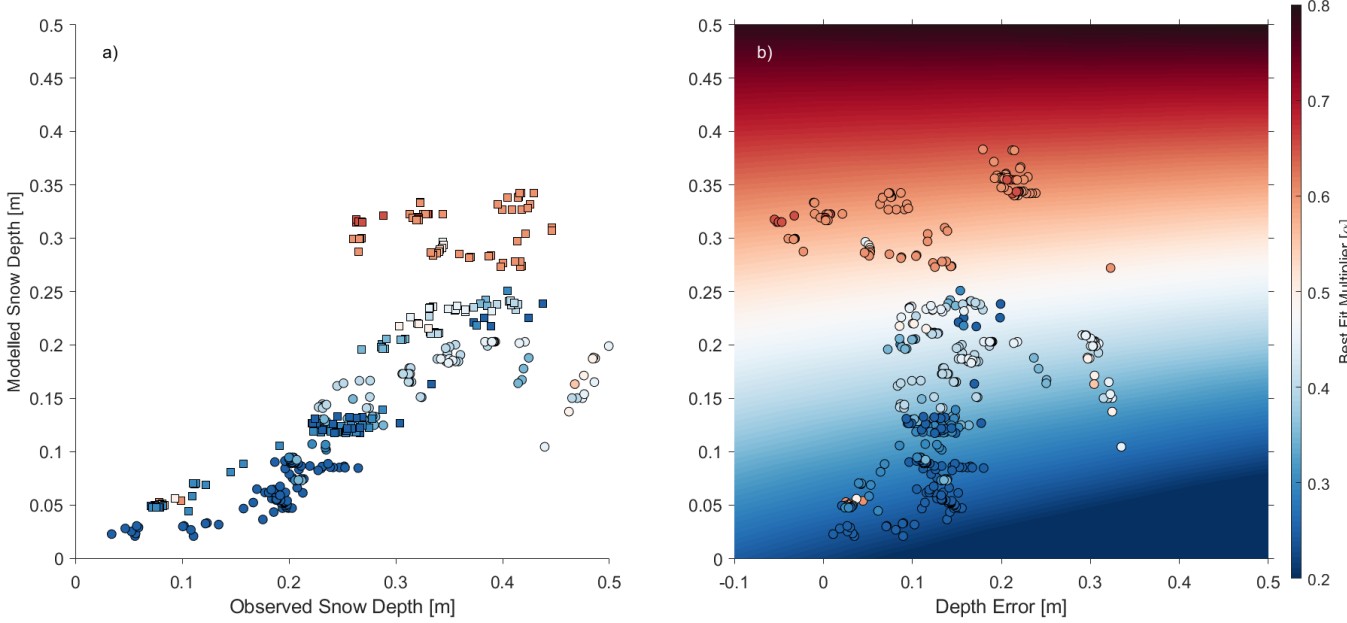

**Figure 8: Influence of observed (x-axis; a) and modelled (y-axis; both) snow depth and snow depth error (x-axis; b) on**
**the best fit correction (colour) at each timestep for both the 2017-18 (circles) and 2018-19 (squares) snow seasons.**

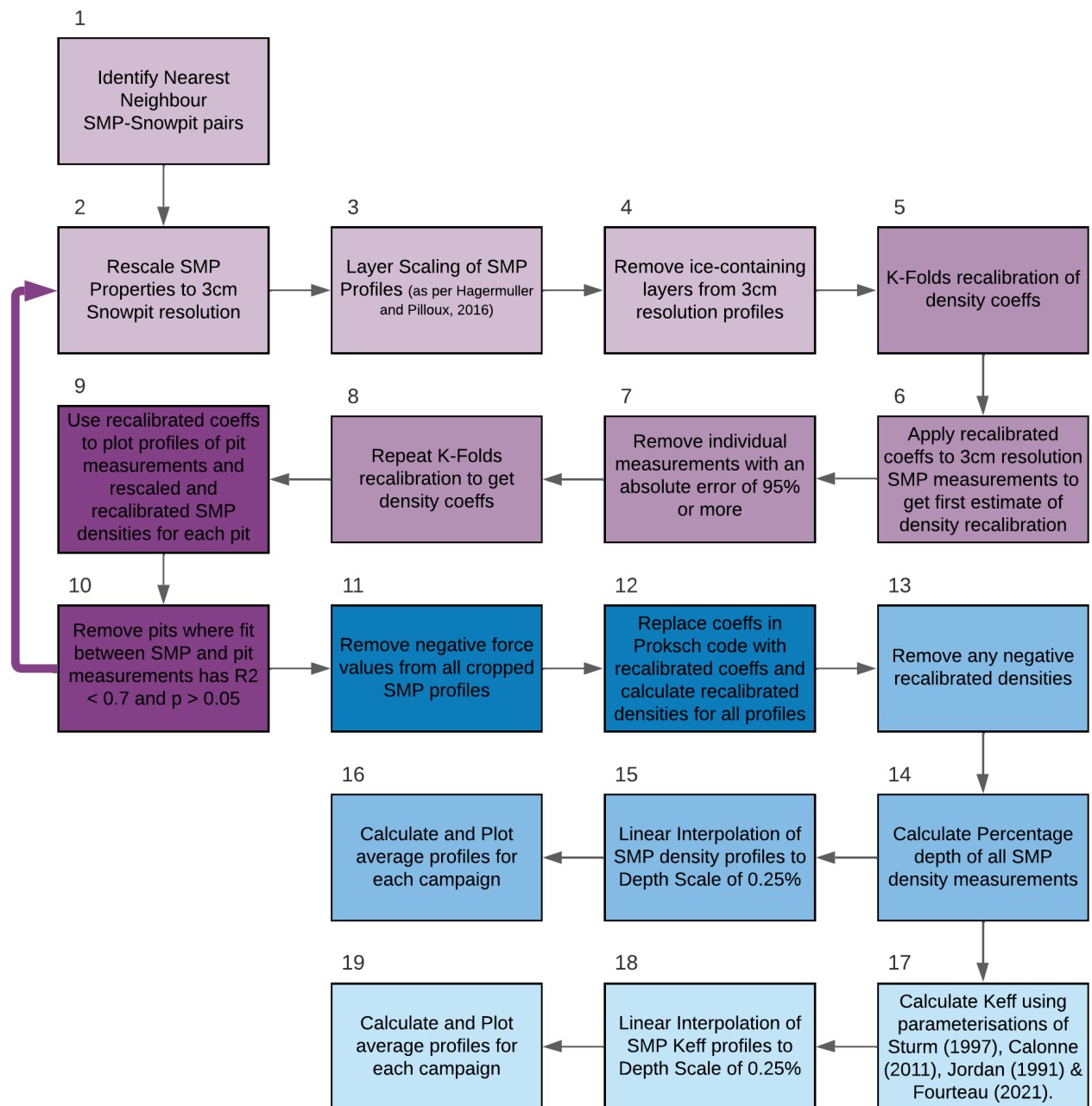

**Figure A1: Recalibration process for SMP densities. Steps 2-9 (purple) mirror the process of King et al. (2020b), with Step 10 providing a quantitative threshold to assess whether the recalibration attempt is successful. Steps 11 to 19 (blue) apply the recalibration to the TVC dataset and derive thermal conductivity profiles from the recalibrated SMP densities.**

| | *a* | *b* | *c* | *d* |
|---|---|---|---|---|
| Proksch (2015) | 420.47 | 102.47 | -121.15 | -169.96 |
| King (2020b) | 312.54 | 50.27 | -50.26 | -88.15 |
| **This Study** | *307.36* | *43.51* | *-38.95* | *-79.36* |

**Table A1: Coefficients used to calculate density from SMP measurements.**


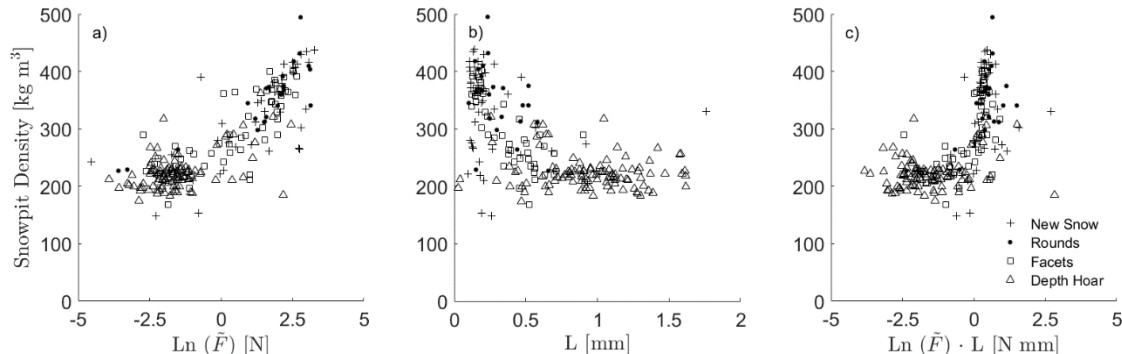

**Figure A2: Relationships between the snowpit densities and SMP microstructural metrics from the paired profiles in the recalibration dataset, after Figure 5 of King et al. (2020b).**


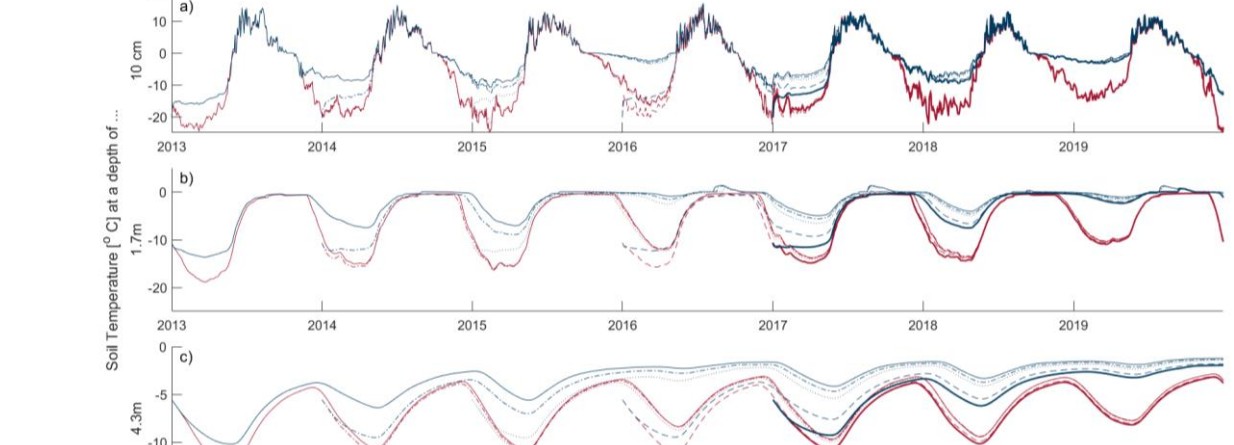

**Figure B1: Soil temperatures at a) 10 cm, b) 1.7 m and c) 4.3 m (third, eleventh and sixteenth CLM soil layers) for varying lengths of model spin-up (line styles; all spin-ups from 1 January, year given in legend), for both baseline ($\alpha = 1$; dark red) and $K_{eff}$ adjusted ($\alpha = 0.3$; navy blue) model conditions.**