# Peer review of "Impact of measured and simulated tundra snowpack properties on heat transfer"

_The Cryosphere, 2021_

## Author Comment (AC1)

**Author Response for** *"Impact of measured and simulated tundra snowpack properties on heat transfer"*, **Victoria R. Dutch et al.**

We would like to thank the editor and both reviewers for taking the time to read and comment on the original manuscript. We provide a response to these comments below. For ease, comments are in black and our responses in blue. All minor grammatical fixes listed under the edits subheading have been changed.

**Reviewer 1:**

*General comments:*

Thermal conductivity of snow is critical and still very challenging to access. Field measurements have been proven to be satisfactory to a certain level but the inclusion of theoretical equation (or assumption) into models is not yet actual. While most of the main parameters have been discussed and tested, and appropriate literature cited, in order to explain the differences between measured and modeled snow KT, the hardness of the snow pack has never been mentioned in the work, neither its permeability. For sure hardness is kind of included if we think of the measurement using micropen, but as demonstrated in the work, such measurements had to be tuned to the Arctic snow pack as originally validated for Alpine snow packs. Vapour kinetics transfer are necessary to form depth hoar, but the original structure of the snow on ground will also affects such transfers.

Thank you for recognizing the importance of properly considering snow physical properties in order to advance process understanding and model parameterizaton of snow thermal conductivity. A novel component to our analysis is the use of SMP measurements to determine high vertical resolution tundra snow density profiles, hence a new approach to the consideration of 'snow hardness' is central to our analysis. While the SMP measurements clearly show the importance of the basal depth hoar layer in influencing the thermal conductivity, this layer is not well captured by land surface models, which is no doubt a key driver of the offset in thermal conductivity values that we illustrate in Figure 6.

The depth proportion of each different layer is indicated on fig 2 but not discussed, as for example to presence of indurated depth hoar in 2019 and not in 2018. Both seasons had some warming events, which seems to not have affected the soil temperature, but what about the snow pack and the formation of internal, even small, ice layers?

A description of the snowpack layering differences is provided in Section 3.2. We have added the following sentence:

"Ice lenses were present in March 2018, but not the 2019 campaigns"

Are there any relation between the wind (speed threshold during storms?), the amount of rain/positive degree days and the errors in term of soil temperature? It is also surprising that no wind data are presented in the climatology...

Figure 7a clearly shows that during the summer period, when temperatures are above zero, and there are PDDs (positive degree days), that the error between modelled and observed soil temperatures is insignificant.
The error in soil temperature during the winter season cannot be correlated with the amount of PDDs during the winter – because there are no PDDs during the period where the soil temperature regime is established (Fig. 1a). It would also not be sensible to attempt to relate summer PDD total

to soil temperature model/observation bias because we'd be doing so with only two datapoints from two snow seasons.
The magnitude of the simulated errors in soil temperature are dictated by the timing, rate, and vertical snowpack structure during the period of snow accumulation. It is possible that rainfall onto the snowpack will alter the properties of the snow itself, but there were no significant rain on snow events during the two winter seasons in this study. Windspeed is important because wind slab properties will influence the thermal conductivity of the snow. However, we lacked reliable wind speed measurements during the winter and so could not consider the role of wind speed on the wind slab properties.

Snow depth has been shown in the work to be the dominant factor for applying the correcting factor, but if one think of snow being deposited as a single homogeneous layer, then only the physical snow parameters are of interest. The amount of snow layer and their internal properties would be then the controlling factor, for example an very hard and dense snow layer or thick ice layer with a low permeability likely to enhance the formation of depth hoar. Such phenomena are likely to be more present in the future as it is expected more extremes events in the Arctic. I do not think it is necessary to change a lot of the work done and am sure all the data are available in the runs done.

A discussion around that topic is to me of interest in order to pave the future work into such topics and go further into the effects of snow heat properties on the soil's ones, as it is unlikely in the coming years we would be able to develop, and run, a full model with a proper physical scheme for snow thermal conductivity, without considering the validation. It is then today only by using such work presented in this paper that we could be able to better understand the soil-snow-atmosphere feedback in the Arctic.

Thank you for your supportive comments! We agree that this is an important area of research.

*Minor comments:*

Line 117: am not sure a value of 2 for k is what was intended to write

Yes, 2 is the intended value. Now written as 2.0 to make this more obvious.

Figures 8 and 9 appear before 5, 6, and 7

Figure 8 was referenced before figure 7 in the original manuscript, so these have now been renumbered. All other figures are now cross-referenced in order.

Figure 3, November appears before March from left to right

From left to right, months appear in chronological order.

**Reviewer 2:**

*General comments:*

the CLM snow module seems from scratch unappropriate to simulate the properties of the Arctic snowpack. This finding is clearly not new ; previous studies, notably Barrere et al., 2017, and others well cited by the authors, have highlighted the deficiences of this kind of models in the context of Arctic snow.
Agreed. This is the motivation for the study. CLM is a widely-used global model (including in the study of Natali et al. (2019) on winter season carbon flux).We address the representation of thermal conductivity in order to improve simulation of soil temperatures, due to the potential implication

erroneous soil temperature simulations may have on estimates of Arctic carbon cycling, rather than solely thinking about how poorly it simulates the structure of Arctic snowpacks.

 In the present version of the manuscript, the description of the snow module of CLM is so light, that it is hard to capture the key features and fonctionning of the model. Typically, are the modifications Van Kampenhout et al., 2017, used in the present study ? It is not clear to the reader. As the CLM snow model is at the core of thisstudy, a more enhanced description of the fonctionning of this model is required in the Methods of the paper : presently the line "Each layer is parameterised using layer thickness, temperature and mass of water and ice, as per Anderson (1976), Dai and Zeng (1997) and Jordan (1991)." (L37 p4) is much too vague as each of these publications contain different variants of constitutive equations. In the present form, it is not possible for the reader to understand which one is used for which process/variable. The constitutive equations for the evolution of layer thickness (hence compaction) and ice/water content or density should be explicited or at least explicitly referenced.

We have substantially enhanced the description of the CLM snow module , which now reads as follows:

"Developments between CLM4.5 and CLM5.0, as outlined in Van Kampenhout et al. (2017) improved the snow scheme in CLM. The version of the model used herein produces a computationally-layered snowpack, with the number of snow layers dependant on the snowpack depth, up to a theoretical maximum of 12 layers (as opposed to the 5 layer maximum in previous versions of CLM). Once the total snow depth exceeds a given threshold, the initial snow layer is subdivided into two layers with equal properties. Snow layer formation continues in this manner as layer thicknesses surpass the prescribed ranges given in Jordan (1991). When a layer divides, the new layer is formed beneath it, rather than new layers being formed at the surface by new snowfall. As this process is not stratigraphically representative, layers are not described by snow type (for example, as per Fierz et al. (2009)), but instead numbered from the snow surface down. Layer thicknesses are also influenced by snow compaction, parameterised followingAnderson (1976). Unsaturated layers may compact due to overburden pressure, the breakdown of new snow crystals or melting, with the thickness of a snow layer a function of the snow thickness at the previous timestep and the rate of compaction. Snow depths below 1 cm are not discretely modelled and are instead combined into the surface soil layer.

Density, thickness and thermal conductivity are output as a daily mean for each layer. CLM calculates snow density as a function of the relative proportions of ice (mass of ice = $m_i$) and liquid water (mass of liquid water = $m_{lw}$), weighted by the snow cover fraction ($F_{sno}$) for each grid cell (Lawrence et al., 2018):

$$\rho = \frac{m_i + m_{lw}}{F_{sno} \times h_{sl}} \tag{1}$$

In practice, due to the adjusted snow cover fraction and as liquid water in the snowpack is zero until the start of melt out, the computed snow layer density simplifies to the mass of ice ($m_i$) divided by the height of the snow layer ($h_{sl}$). Fresh snow density is influenced by both temperature and wind, with the density of the snowpack allowed to evolve as a result of the compaction processes outlined above. CLM does not allow for temperature-gradient metamorphism, and thus does not represent the development of depth hoar layers (Van Kampenhout et al., 2017).

The computed snow layer densities are then used to calculate snow layer effective thermal conductivities ($K_{eff}$), as per Jordan (1991):

$$K_{eff} = K_{air} + \left(\left(\left(7.75 \times 10^{-5} \times \rho\right) + \left(1.105 \times 10^{-6} \times \rho^2\right)\right)\left(K_{ice} - K_{air}\right)\right) \qquad (2)$$

Values for $K_{ice}$ and $K_{air}$, the thermal conductivities of ice and interstadial air, are given in Lawrence et al. (2018). Snow (and soil) temperatures are defined for the midpoint of each layer at an hourly resolution, with the soil column consisting of 25 layers of increasing thickness (down to a depth of 49 m). Despite the simplicity of the snowpack scheme included in CLM, previous evaluation of snow heat transfer in CLM4.0 (Slater et al., 2017) suggests this modelling framework should perform well."

CLM relies on the parametrization of snow thermal conductivity by Jordan, 1991, as recalled in eq 2. However, the authors chose to derive their conductivity profiles from observations using the Calonne and the Sturm parameterization, and not the one by Jordan, which would have allowed a much more direct comparison to CLM results (eg Fig 6). How do you justify this choice, how does the Jordan, 1991 parameterization compare to the others ? Is the effective thermal conductivity mentionned by Jordan et al., 1991 (p18) and accounting for heat transport through conduction and vapor diffusion, effectively used in CLM (and not just the k_snow) ? This should be explicited.

This is an excellent point. As suggested, we have now calculated thermal conductivity from the SMP using the Jordan approximation. The results of the Jordan parameterisation are very similar to those from that of Calonne (2011). Sections 3.2 & 3.3, and Figures 4 and 6a have now been updated to include this.

[Figure]

Figure 4: Median thermal conductivity profiles (lines) and interquartile range (shaded areas) approximated from SMP densities, using the parameterisations of Calonne et al. (2011) in black/grey, Jordan (1991) in red, and Sturm et al. (1997) in blue.

The following is more a minor comment : please also beware of the use of the term "effective conductivity". Following Calonne et al., 2011 and in a material science perspective, effective conductivity refers to the conductivity of the ice-air|liquid water mixture that constitutes the snow material, while individual components like ice or air, have just a thermal condictivity. In other studies, and often in land-surface modelling as in Jordan et al., 1991, "effective conductivity" refers

to the additional inclusion of latent-heat exchange within the conductivity used in the Fourier heat transfer equation. So the use of Keff L149 p4 is appropiate, while its uses L151 p4 are not.

The word "effective" has been removed from what was L151.

As a general question, is there any perspective to generalize the bias correction factor designed here, and have in the future global CLM runs using this usefull correction ? This would strongly enhance the impact of the work carried out in this study.

This is a great question. We are currently working on this to assess near-surface soil temperatures and their implication on soil/microbial respiration fluxes in CLM. Running CLM across a larger number of sites will help in understanding how to generalise correcting snow conductivity. We have also included a paragraph on determining where it is appropriate to make such corrections:

"However, these approaches to improving snow thermal conductivity may not be globally appropriate, especially in climates where temperature gradients through snow are insufficient to create major depth hoar layers and where compaction is the dominant process controlling vertical profiles of snow density. The Sturm et al. (1997) parameterisation demonstrates transferability between tundra sites, having been derived from thermal conductivity measurements on the North Slope of Alaska and successfully applied to both CLM and SMP measurements at TVC. Regardless of approach, the empirical correction factor α or the Sturm et al. (1997) thermal conductivity parameterisation, these changes to the model are most applicable where snowpack structure is considerably influenced by depth hoar, as can be approximated by grid-cell plant functional type or climatology (Royer et al., 2021a; Sturm and Liston, 2021)."

*Minor comments:*

P3 L110-113 : the way it is formulated gives the impression that the snowpack always entailed the 4 disctinct layers mentionned. I imagined that in practise, more complex layerings were sometimes encountered. Maybe just reformulating saying "Stratigraphic information profiled in each snowpit (n = 115) was used to assign layer types to the measured densities (Fierz et al., 2009), among four different layer types : surface snow, wind slab, indurated hoar and depth hoar. This was made to assess spatial variability in the thickness and properties of different snowpack layers."

Sentence now reads "Stratigraphic information profiled in each snowpit (n = 115) was used to assign one of four different layer types (surface snow, wind slab, indurated hoar and depth hoar) to the measured densities (Fierz et al., 2009) in order to assess spatial variability in the thickness and properties of different snowpack layers."

P3 L112-119: It is hard to understand the nature and impact of these adjustement without being quite familiar with CLM. I would suggest, if this PTCLM version is not published/referenced anywhere else, to just mention "adjustements" here and maybe detail them a bit more explicitly in an Appendix, so that the curious reader may dig full, explicit information from there.

Additional information has now been added, and the paragraph now reads as follows:

"The Community Land Model v5.0 (CLM; Lawrence et al. (2019)) is the land surface component of the Community Earth System Model v2.0, which can be run at a variety of spatial scales. In this study, 1D "point mode" (a 0.1° x 0.1° grid cell) CLM (PTCLM; Kluzek (2013)) simulations were centred at the location of the TVC station. Minor adjustments were made to the model in order to better emulate snow accumulation and melt at the point scale; the snow accumulation factor was

increased (Swenson and Lawrence, 2012) from 0.1 to 2.0 and the standard deviation of elevation set to 0.5 m after Malle et al. (2021; Figure S4). These adjustments limit the period of fractional snow cover, so that PTCLM represents a binary state of snow presence or absence over a flat surface. PTCLM simulations were run from August 2017 to August 2019, with model spin-up from January 2013. Spin-up was necessary in order to allow soil temperatures to equilibrate. Variation between model runs with the same parameterisation after more than 2 full years of spin-up is limited to ~ 1 $^{\circ}$C throughout the top 5 m of the soil column. The impact of spin-up on soil temperature is further discussed in Appendix B"

P4 L126 : was a debiaising of ERA5 or adaptation to the local conditions maybe required?(see : https://doi.org/10.5194/tc-2021-255 )

The following information has been added;
"Gapfilling was only required for measurements of incoming longwave and shortwave radiation, and comparison of observations and reanalysis data showed an offset of less than 60W m$^{-2}$. Bias correction of reanalysis data was not undertaken due the small size of this offset."

P5 L 171 : "slower rate of soil freezing" : a delay is visible, but not so clearly a slower rate. Could you justify this more with the observations ?

The words "rate of" have been removed from this sentence.

P5 L176-178 : "anomalously warm mid-winter air temperatures... had little influence on the soil temperature profile" : the March 2019 warming is visible though on the soil temperatures, isn'it ?

Sentence was meant to imply only a limited impact, but has now been rewritten and quantified to make this clearer. Now reads: "Anomalously warm mid-winter air temperatures that approached 0 $^{\circ}$C (22 December 2017 and 9 February 2019) or exceeded 0 $^{\circ}$C (18 and 31 March 2019, with a rain-on-snow event occurring on the latter of these dates) had a muted influence on the soil temperature profile (Fig. 1d). For example, as air temperature increased by 25 $^{\circ}$C between 14 and 31 March 2019, 20 cm soil temperatures increased by 2.5 $^{\circ}$C, as air and soil temperatures were decoupled due to snow insulation."

P6 L 211: "Temperature-gradient" metamosphism should be specified here (as wet-snow metamorphism would have different consequences !).

Added. Sentence now reads "The density of this layer became more variable as each winter progressed due to the competing processes of wind compaction (increasing density) and temperature-gradient metamorphism (decreasing density)."

P6, paragraph 3.2 : I think that a line on the increase in density and effective thermal conductivity towards the bottom of the snowpack should be added, to complete the description of the profiles. I noted that possible reasons for this are given in the Discussion, and it could be referred to here.

Description of the density profiles in the results section is now as follows:
"Snowpacks in all three campaigns (Fig. 2b-d) had a very thin surface snow layer (composed of recent snowfall), with low near-surface snow densities (< 300 kg m$^{-3}$) rapidly increasing in the top 5 % of the snowpack. A higher density (~ 320 kg m$^{3}$) wind slab layer was evident between 5 - 30 % of normalised depth from the snow surface. The next ~ 10 % of the profile was a transitional section

where density decreased by about 100 kg m$^3$. The lowest ~ 60 % of the profiles is dominated by a lower density (~ 230 kg m$^3$) depth hoar layer, the density of which increases slightly towards the base of the snowpack"

P6 L237 : "Whilst the absolute number of simulated snow layers is plausible," : these layers have no physical meaning in the model, this is well explained in the methods. Therefore the remark is in my opinion unappropriate, or it should be specified w/r to what this is plausible.

Good point. Removed. This sentence now reads: "The physical properties of the simulated snow layers do not correspond to observations, with the number and thickness of snow layers only a function of overall snowpack depth."

P7 L250: " with the median thermal conductivity using the Calonne approximation still notably lower". The difference between simulations (0.3 Wm-1 K-1) and the median thermal conductivity using the Calonne approximation (0.25 Wm-1 K-1) is not so high, given the uncertainties attached to thermal conductivity estimations, and their range of variations. I suggest to suppress "notably".
The word "notably" has been removed.

P7 L 264 : what is the "maximum duration of simulated snow cover" ? (the period with continuous snow cover on the model ? )

This has now been edited to read "the maximum annual duration of continuous simulated snow cover (15 Sept – 31 May; RMSE = 5.8 ℃)."

P7 L 266-272: wouldn't it be easier to say that density was multiplied by a corrective factor α prior to the calculation of Keff ?
Snow density and thermal conductivity are calculated within different modules within CLM. In order to conserve mass and prevent model instabilities, the multiplier was only included in the module where Keff is calculated – applying the multiplier to the simulated density for the calculation of Keff did not result in a change in snow density at this or any following timesteps.

P7 L 274 : " between the interquartile range of observed values shown in Table 2" : for a given snow type of for all ? If for all, the range should be specified as it is not in Table 2 if I am not mistaken.
The range of values used spanned the interquartile ranges of all snow types. Changed to "between the interquartile range of observed values for all snow types (73 – 365 kg m$^3$)."

P7 L 284-286 : It is really not clear to me how this linear regression was performed, could you give more details ? (also regarding the temporal sampling used)
More detail about the temporal sampling has been added. Sentence now reads:
"A multiple linear regression was undertaken to quantify the influence of snow depth and snow depth error on the value of the best fit correction factor, for the period from snow onset to the start of simulated snow melt-out (when the simulated snow cover fraction was equal to one)."

P9 L337 and 361 : "vital" seems a bit strong and out of place in this context
The word vital has been removed.

P9 L 346-347: maybe also mention here that the insulating effect of snow somehow saturates after a certain snow height has been reached, impliying a stronger sensitivity of soil thermal regime to snow depth in the early winter close to snow onset, when the snow cover is very thin.
The following sentence has been added to the end of the section:

"The soil thermal regime is also more sensitive to snow depth at the start of the season as snow depths are lower and have not yet reached a point where their insulative capacity has become saturated (Zhang, 2005; Lawrence and Slater, 2009; Slater et al., 2017)."

P9 L356: Comparing this point simulation with adjusted Keff, to CMIP5 models evaluated globally (if I am not mistaken..) is actually very unfair to the CMIP5 models ! The string differences in setup should be mentionned to balance this statement.

The following edit has been made to balance the comparison to the CMIP5 models:
"Ignoring interannual variability and calculating one SHTM value (SHTM = 0.733) for the entire model run appearing to show better model performance than 8 of the 13 CMIP models compared in Slater et al. (2017) and suggesting a more accurate simulation of soil temperatures from the baseline model run than seen in Fig. 7a. However, we note the role of different site conditions and model configurations on intercomparisons, a important caveat when comparing our single point simulation to the global CMIP5 simulations."

P9 L354-358: in relation to one of my major comments : did you tests other parameterizations than Jordan, 1991, for Keff in CLM, for instance the Sturm ? (I do not know how it positions w/r to the Jordan, but this should be mentionned if likely to also yield improvements).
We have now re-run the model with the Sturm parameterisation of thermal conductivity, as well as the default (Jordan) parameterisation. We have added the following paragraph to Section 3.4;

"Additionally, simulations were also undertaken where the default parameterisation for snow thermal conductivity (Eq. 1; Jordan (1991)) was substituted for that of Sturm et al. (1997):

$$
\begin{aligned}
\rho \leq 156 & \quad K_{eff} = 0.023 + 0.234\rho \\
\rho > 156 & \quad K_{eff} = 0.138 - 1.01\rho + 3.233\rho^2
\end{aligned}
\tag{5}
$$

The Sturm parameterisation resulted in lower simulated thermal conductivities (Fig. 6b) and closer temperatures to observations (Fig. 7b; RMSE = 2.5°C). Soil temperatures in 2017-18 were still too cold regardless of parameterisation used, likely due to model underestimation of snow depth (Fig 7c)."

We have also added an additional paragraph to Section 4.3 discussing the implications of these results:

"Changing the parameterisation of snow thermal conductivity in CLM from that of Jordan (1991) to that of Sturm et al. (1997) improves the simulation of both snow thermal conductivity values and underlying soil temperatures. Similar issues in snowpack and soil temperature simulations have been found for simulations of Arctic snowpacks using other models, i.e., Crocus, SNOWPACK, ISBA-ES (Barrere et al., 2017; Domine et al., 2016; 2019; Royer et al., 2021b). Use of the Sturm et al. (1997) thermal conductivity parameterisation also improved soil temperature simulation in Crocus (Royer et al., 2021b), with a RMSE of 2.5 °C for soil temperatures from Crocus and CLM. However, these approaches to improving snow thermal conductivity may not be globally appropriate, especially in climates where temperature gradients within the snowpack are insufficient to create major depth hoar layers and where compaction is the dominant process controlling vertical profiles of snow density. The Sturm et al. (1997) parameterisation demonstrates transferability between tundra sites, having been derived from thermal conductivity measurements on the North Slope of Alaska and

successfully applied to both CLM and SMP measurements at TVC. Regardless of approach, the empirical correction factor α or the Sturm et al. (1997) thermal conductivity parameterisation, these changes to the model are most applicable where snowpack structure is considerably influenced by depth hoar, as can be approximated by grid-cell plant functional type or climatology (Royer et al., 2021a; Sturm and Liston, 2021)."

This sentence has been added to the final paragraph of the conclusion (Section 5):

Alternatively, different parameterisations of snow thermal conductivity also improve simulation of soil temperatures, with that of Sturm et al. (1997) more appropriate for Arctic snowpacks than that of Jordan (1991) used by default in CLM.

And we have also added the following to the abstract:

"The use of an alternative parameterisation of snow thermal conductivity also improved simulations of wintertime soil temperatures (RMSE = 2.5 °C)."

P9 L363-365: "Larger values of the correction factor are needed to replicate observed soil temperatures later in the winter season, as errors in simulating earlier season snow depth are additive, leading to larger discrepancies for both snow depth and soil temperatures" : Could you specify to which specific Figure or result statement this assessment relate ?

A reference to Figure 7 has been added to this sentence.

P9 L367 : "This bias compensation between underestimates of snow depth and overestimates of snow thermal conductivity": I had rather say bias compensation between underestimates of snow depth and UNDERestimates of snow thermal conductivity (?) please correct me if I am wrong.

Corrected.

P10 L 380 : " Reducing simulated thermal conductivity by 80 % (α = 0.3) produces changes in soil temperatures approximately equivalent to the impact of changing depth hoar fraction from 0 to 60 % (Zhang et al., 1996), suggesting the inclusion of vapour transport in the snowpack is at least equally important as values of snow thermal conductivity in accurately simulating wintertime soil temperatures. ". The sentence is ambiguous and the message hard to understand (not sureI understood properly). The inclusion of vapour transport in the snowpack will change the thermal conductivity of the snowpack by two ways : i) because this will form depth hoar with lower Keff ; ii) because of vapour transprt induced heat transport. Anycase the inclusion of vapour transport in the snowpack, means different values for snow thermal conductivity. I think the sentence should be rephrased.

This sentence has been removed.

P9 L381-384: Honestly, the explicit inclusion of vapour transport within the snowpack in the snow modules of land surface models is a (very) long way from now. I am unaware of very concrete plans in that direction for CLM, but please correct me if they exist. I am much more confident that physically representative approaches, (like Royer et al., 2021), though not physically explicit, will be first used, and this would be the short to mid term perspective.

To my knowledge, no such plans for CLM are currently in the works. We have edited this section to refer to such approaches.

"Further improvements in SHTM in future iterations of CLM will require a physically representative approach to snow density and thermal conductivity through explicit inclusion of vapour transport within the snowpack, currently under development in stand-alone snow microphysical models (Fourteau et al., 2021; Jafari et al., 2020; Schürholt et al., 2021). However, this presents computational and mathematical challenges, as outlined in Jafari et al. (2020). The inclusion of physically representative parameterisations of snow properties in land surface models, such as that of Royer et al. (2021b) where the densities of lower snow layers are not allowed to exceed a maximum observation-based threshold, are more likely in the near future than the explicit representation of snowpack vapour transport. Meanwhile, both the empirically derived scaling factor or the substitution of the Sturm et al. (1997) thermal conductivity parameterisation provide a computationally efficient compromise, reducing both the value of $K_{eff}$ and the cold bias of simulated wintertime soil temperatures considerably (RMSE reduction of 3.7 °C for $\alpha$ = 0.45, RMSE reduction of 3.3 °C for the Sturm parameterisation)."

P11 L448: Figure A2c does not explicitly show that the relationship between FÌ$f$ and L is not heteroscedastic; it is rather the combination of all figures from this panel (?)
This sentence now refers to Figure A2, rather than just the last panel specifically.

Fig 1 is hard to read, maybe consider having a grid (making date correspondances easier to follow), use backgroun color instead of the color of very small histograms for precipitation phase ; maybe also enhance y-axes sizes and line width. Plus, the caption contains some errors, please check.

Figure 1 has been updated as below, white space between plots has been removed and the line widths increased. Caption has also been revised slightly.

[Figure]

Figure 1: Meteorological and soil conditions at Trail Valley Creek from 1 August 2017 to 31 August 2019; a) 2 m air temperature, b) precipitation: snow (purple) and rain (green), c) snow depth, d) soil temperatures at depth of 5 cm (blue) and 20 cm (orange) and e) volumetric soil water content at 5 cm (blue) and 20 cm (orange) depths.

Fig 6 : nice figure !

Thanks! The figure has been updated to include SMP-derived Keff from the Jordan parameterisation (Red dashed line, top plot) and the use of the Sturm parameterisation in CLM (Thick blue line, bottom plot):

[Figure]

The caption now reads:

"Figure 6: a) Histograms of measured and simulated thermal conductivity from March 2018 and 2019 sampling campaigns, b) Sensitivity testing of simulated thermal conductivities for the same time period using both the default CLM snow thermal conductivity parameterisation and the Sturm (1997) parameterisation. Note the different scales on the y-axes."

Fig 8 is mentionned in the text before Fig 7, they should be inverted. Could you thicken the lines a little bit (should be feasible when reducing the y range) – the figure is quite hard to read.

Numbering of figures 7 and 8 has been reversed as detailed in response to Reviewer #1 above. An additional panel has been added to Figure 7, as shown below:

[Figure]

Figure 7: a) Timeseries of 10cm soil temperatures timeseries for $K_{eff}$ sensitivity tests compared to field measurements, b) Timeseries of simulated 10cm soil temperature timeseries using both Jordan (1991) and Sturm et al (1997) parameterisations of snow thermal conductivity,
c) Observed and simulated snow depths for the same time period.

The line weight on the new Figure 8 has been increased, as shown below:

[Figure]

Figure 8: Changes to model performance when adjusting snow thermal conductivity, measured using RMSE of 10cm soil temperatures (solid lines) and the Snow Heat Transfer Metric (SHTM – dashed lines) for the 2017-18 (black) and 2018-19 (green) snow seasons.

---

## Referee Report (RR1)

**Review of the revised version of "Impact of measured and simulated tundra snowpack properties on heat transfer"**

**General evaluation**

These general comments were written after the subsequent specific comments, I apologize for the few repetitions.

This paper in fact reports two somewhat independent topics. The first one is the determination of vertical thermal conductivity profiles of snow near TVC using density profiles determined from SMP profiling. Density is derived from SMP data using an algorithm developed from snow on sea ice, if my understanding is correct. The second one is to compare these thermal conductivity profiles to those simulated by CLM. As expected from any snow model used in the Arctic, CLM does not perform well. It simulates much too cold temperatures and the authors discuss various adjustements to make the model better fit the data.

The experimental part is interesting and there is some novelty to it. There are however a number of weaknesses. First, the thermal conductivity is derived from density using various equations that do not agree with each other. Second, while I am willing to believe that the algorithm used to convert SMP data to density is good, it seems to have been validated for snow on sea ice, which is somewhat different from snow at TVC where vegetation impacts snow properties. Third, in the discussion, the authors mention an artefact where the penetrating SMP rod may have damaged the fragile depth hoar, causing an artefact with lower intermediate density and increased lower density. This third issue is very important and must be mentioned in methods so that the reader has it in mind while evaluating the experimental part. I detail this point below. So, while SMP clearly allows obtaining much more data with high vertical resolution, its limitations and potential artefacts must be stressed, with the possible conclusion that it is not (very) suitable for snowpacks with very fragile basal layers. Or that the wind slab must be removed to measure the depth hoar? In any case, I feel the SMP data deserves publication if the caveats are clearly stated.

Another issue with the derivation of thermal conductivity (TC) profiles from SMP data is the choice of the density-TC relationship. First, the authors must remain aware that this is far from a bijective relationship and that there is enormous scatter in this relationship. For example, in (Sturm et al., 1997) there is sometimes a factor of 5 variation in TC for a given density value. The use of such an equation may therefore bear some implicit very high uncertainty and error. Second, there are large differences between the various relationships used, so which one is correct, if there is one, and what is the error induced by these relationships?

I would like to comment on the various relationships used. The equation of (Jordan, 1991) is based on that of (Yen, 1981) which is in fact a compilation of previous works using a variety of methods. (Sturm et al., 1997) use only the needle probe method, but there are doubts on its reliability (Riche and Schneebeli, 2013) and this is not just due to anisotropy issue as stated by the authors. (Fourteau et al., 2022) have very recently made a detailed theoretical and experimental study of the needle probe method and demonstrated that the aproximate equation used by (Sturm et al., 1997) to derive thermal conductivity from heated needle probe data is not correct, leading to a negative artifact. Briefly, it is not valid for the short heating

times used as some neglected terms in the full equations are in fact not negligible. (Fourteau et al., 2022) proposed an algoritm to correct existing needle probe data. In any case, I think the equation of (Sturm et al., 1997) is incorrect, and the fact that it produces values lower that the other parameterizations confirms this negative artefact. I very strongly recommend to stop using this equation. The equation of (Calonne et al., 2011) is based calculations from tomographic images. It is rigorous, however, it minimizes water vapor effects because is postulates slow surface kinetics. In other words the accommodation coefficient of water vapor on ice takes here a low value, so that latent heat fluxes are minimized and probably underestimated. More recently (Fourteau et al., 2021) used a similar approach but postulated fast surface kinetics, leading to slightly higher values than (Calonne et al., 2011). In conclusion, the equation of (Sturm et al., 1997) is incorrect and should just not be used. The equation of (Jordan, 1991) is based on ancient measurements using a variety of poorly documented techniques and I feel it has limited reliability. The equation of (Calonne et al., 2011) is arguably correct but makes the debatable postulate that surface kinetics is slow, while (Fourteau et al., 2021) make the more reasonable (in my opinion, but I am a coauthor of that paper) postulate of fast surface kinetics. My biased recommendation is therefore to use the equation of (Fourteau et al., 2021) and in any case not to use the equation of (Sturm et al., 1997). I recommend not to use different equations as an error compensation trick in CLM.

In fact, I am not too thrilled by the modeling part. Essentially this just says that a simple snow routine from a more complex model just does not work in the Arctic. It is interesting to confirm that the CLM snow scheme is deficient, but I do not think all the error compensation tricks used by the authors, and the lengthy discussion on parameter adjustments, have much interest. Figure 7 shows that for the first year, $\alpha$ =0.4 must be used while for the second year $\alpha$=0.6 is better. So, the adjustment parameter changes from year to year, and is probably different for other sites, so that CLM has no predictive value when it comes to Arctic snow and ground temperature. We already know, I as well as all the authors, that even the currently most sophisticated snow model do not work in the Arctic, so no one expects the simpler CLM scheme to perform any better. The authors just need to show Figure 7, demonstrating the lack of predictive value of CLM and therefore its much reduced interest for predicting snow properties and soil temperature in the Arctic. They then could just discuss that implementing the missing process, upward vapor transfer, is too complex, and that other approaches they wish to propose must be envisaged. I Therefore think Figures 1, 2, 3, and 7 are interesting. I strongly recommend that the authors consider removing the other Figures, or at least most of them, and reduce the modeling text by at least 50%, probably more. Please also consider my comment of Figure 5 below.

Another general comment is that there is a serious lack of attention to detail in the writing and presentation. This is surprising given that "All authors were involved in reviewing and editing prior to submission" and that the authors include a large number of high-profile esteemed and highly respected senior researchers. For example (just one, and I will not edit for typos, the authors can do it), all of these authors think it is fine to write "Snow has a low thermal conductivity, typically in the range $0.01 - 0.7$ Wm$^{-2}$ K$^{-1}$". So snow can have a thermal conductivity less than half that of air? And for brevity I will refrain from commenting the 0.7

value. In any case, I will do my best to write a hopefully constructive review. Let the authors do their best to write a paper with attention to detail.

**Specific comments**

Line 44. I am not sure what the authors mean by "indurated depth hoar". They seem to have a definition different from mine (Domine et al., 2016b) and from Sturm's (Sturm et al., 2008). Sturm and I have the same definition, since Sturm introduced me to indurated depth hoar on the Alaska north slope in 2004. I would think the authors would have a similar definition since Chris Derksen has done much Arctic snow field work with Sturm. However their stratigraphy does not show any indurated depth hoar, but faceted crystals. This is very confusing. I would also think the lower layer, with densities reaching 300 kg m$^{-3}$, would often be actual indurated depth hoar, possibly formed from melt-freeze layers. In any case, I suggest the authors realize that the classification of (Fierz et al., 2009) was made by avalanche experts for avalanche motivations. It is almost exclusively based on observations in Alpine snow and is largely inadapted to Arctic snow. I discussed this with Charles Fierz and he had never seen indurated depth hoar. I think Arctic snow researchers should use symbols adapted to their problem. I have proposed symbols for indurated depth hoar and indurated faceted crystals (stage prior to indurated depth hoar) in (Domine et al., 2016b) and (Domine et al., 2018) (already cited by the authors) and in other papers. Why not popularize these symbols, adapted to Arctic snow, which by the way is much more important area-wise than alpine snow ? It would spare us these inevitable inconsistencies between text and Figures.

Line 150. What is $h_{sl}$?

Line 185-190 are not necessary. These are very well-known considerations. In general section 3.1 can be greatly condensed.

Figure 2: Faceted crystals? Columnar DH? Please clarify symbols and make them consistent with earlier parts of paper. Plus, again, Fierz 2009 inadapted to Arctic snow. How about showing fall 2018 data? Fall data are in general scarce amnd therefore valuable.

Table 2 lacks detail and does not correspond with stratigraphy of Figure 2. There seems to be a dense basal layer, perhaps indurated depth hoar, in the lower 10-20%. Then the next 20-60% seem to have a homogeneous low density typical of columnar depth hoar. Therefore, these 2 layers should be separated in Table 2. Just having one DH layer is not consistent with data.

Line 214. Please descrive indurated DH. Written as faceted crystals in Figure 2.

Line 221. Ice lenses not shown in stratigraphy.

Line 235. Please check grammar

Line 257. Please be consistent with snow layers. Figure 2c and d mention 5 layers. Here just 3.

Line 265. 0.344 is 4 times as large as 0.08, not 3. Sturm is not an approximation, but an equation, or a parameterization.

Line 268. Most of Sturm's measurements were in fact in the Boreal forest of interior Alaska, where the mostly depth hoar snowpack layers have very low thermal conductivities. Many measurements were also from tundra snow but what the authors say is incorrect.

Line 281. Probably unnecessary explanation.

Figure 5. The graphs may be easier to visualize if the grey and black colors were swapped. In any case, I am not sure about the utility of panels b-d and in fact I am surprised by how CLM seems to work. The origin seems to be the top rather than the base of the snowpack, even though the snowpack forms from the bottom up, as the authors know. Following density from the top then does not really correspond to anything physical, as a given snow layer is not monitored. I do not have time to get into the methods and architecture of CLM, which is totally unknown to me, but it seems very strange. Since it does not seem to take into account actual processes, it seems that having such a model fit data will result in mandatory adjustments on a case by case basis, with no hope of ever having any predictive value. From what little I understand, I therefore wonder whether there is actually any hope of ever getting any reliable snow simulations from CLM. I'll be more than happy to be proven wrong.

Line 293. Why just mention 2018-2019? Is not it interesting to realize that in 2017-2018, $\alpha$ =0.4 works best? A single value cannot simulate both years. And therefore, expectedly, different thermal conductivity parameterizations have to be used for each year. This may be stressed.

Line 296. I do not understand (0.3 ≥3≥ α ≥ 0.5555)

Line 316. Sturm's parameterisation works best, but that parameterization is wrong. This is just an error compensation game. The model is wrong, and this is compensated by a wrong thermal conductivity parameterization. And the compensation is different for each year, meaning that the model has no predictive value.

Lines 326-328. This comparison is very misleading. The authors compare their indirect estimation of thermal conductivity based on an indirect estimation of density to actual measurements of snow thermal conductivity. Furthermore, the measurements of (Domine et al., 2015; Domine et al., 2016b) and of (Morin et al., 2010) are continuous season-long time series, so that the focus of those papers are on time-variations, while the authors'work is on height variations. By the way, vertical profiles of thermal conductivity measured by (Gouttevin et al., 2018) and by (Domine et al., 2012; Domine et al., 2016a) is closer to 5 cm resolution than to 10 cm.

Lines 330-344. This discussion is interesting but I think it should already be stated in the methods section. I have been wondering about this high density basal layer, thinking it may reflect rain-on-snow in the fall but only now do I realise it is probably just an artifact! I have been misled in my understanding of the data all along! So please shift this up. And by the way, why did not the authors perform SMP measurements with the wind slab remove to test for the actual impact of this artefact, since they must have been aware of it while making the measurements.

Lines 345-352. Continuous vertical profiling is indeed very nice and is clearly a significant improvement over discrete layer sampling. However, is SMP suited to Arctic snowpacks, with a

basal depth hoar layer that can be extremely fragile and collapse at the slightest touch? (see details in (Domine et al., 2016b). The authors seem aware of this problem, and this clearly limits the interest of SMP for some Arctic snowpacks. Not all, I agree, since very windy areas such as Barrow and polar deserts seldom have very fragile depth hoar. This should also be discussed. Furthermore, SMP is blind sampling, since a snowpit is not dug in most cases as this would cancel the benefit of the technique. Therefore, artefacts due to soft layers would be undetected. In conclusion, while I do see the benefit of SMP in some cases, the authors may wonder whether a low resolution reliable manual density profile is  better than a high resolution profile with potential and unverifiable artefacts.

Line 355. Alpine snow does have a density profile as simulated by CLM, but not the taiga snow, which by the way is not discussed by the references cited. In the boreal forest (for some reason, I refrain from using Russian words these days…?) the profile may be as in CLM at the beginning of the season but by the end of winter it is either flat or with lower basal density because of the upward vapor flux. See e.g. (Taillandier et al., 2006).

Line 361-362. "such as the snowpack vapour kinetics necessary to form depth hoar." The term "vapor kinetics" is unclear. Use flux, and more accurately upward flux.

**References**

Calonne, N., Flin, F., Morin, S., Lesaffre, B., du Roscoat, S. R., and Geindreau, C.: Numerical and experimental investigations of the effective thermal conductivity of snow, Geophys. Res. Lett., 38, L23501, 2011.
Domine, F., Barrere, M., and Morin, S.: The growth of shrubs on high Arctic tundra at Bylot Island: impact on snow physical properties and permafrost thermal regime, Biogeosciences, 13, 6471-6486, 2016a.
Domine, F., Barrere, M., and Sarrazin, D.: Seasonal evolution of the effective thermal conductivity of the snow and the soil in high Arctic herb tundra at Bylot Island, Canada, The Cryosphere, 10, 2573-2588, 2016b.
Domine, F., Barrere, M., Sarrazin, D., Morin, S., and Arnaud, L.: Automatic monitoring of the effective thermal conductivity of snow in a low-Arctic shrub tundra, The Cryosphere, 9, 1265-1276, 2015.
Domine, F., Belke-Brea, M., Sarrazin, D., Arnaud, L., Barrere, M., and Poirier, M.: Soil moisture, wind speed and depth hoar formation in the Arctic snowpack, J. Glaciol., 64, 990-1002, 2018.
Domine, F., Gallet, J.-C., Bock, J., and Morin, S.: Structure, specific surface area and thermal conductivity of the snowpack around Barrow, Alaska, J. Geophys. Res., 117, D00R14, 2012.
Fierz, C., Armstrong, R. L., Durand, Y., Etchevers, P., Greene, E., McClung, D. M., Nishimura, K., Satyawali, P. K., and Sokratov, S. A.: The International classification for seasonal snow on the ground UNESCO-IHP, ParisIACS Contribution N°1, 80 pp., 2009.
Fourteau, K., Domine, F., and Hagenmuller, P.: Impact of water vapor diffusion and latent heat on the effective thermal conductivity of snow, The Cryosphere, 15, 2739-2755, 2021.
Fourteau, K., Hagenmuller, P., Roulle, J., and Domine, F.: On the use of heated needle probes for measuring snow thermal conductivity, J. Glaciol., doi: 10.1017/jog.2021.127, 2022. 1-15, 2022.
Gouttevin, I., Langer, M., Löwe, H., Boike, J., Proksch, M., and Schneebeli, M.: Observation and modelling of snow at a polygonal tundra permafrost site: spatial variability and thermal implications, The Cryosphere, 12, 3693-3717, 2018.

Jordan, R.: A One-Dimensional Temperature Model for a Snow Cover. Technical Documentation for SNTHERM.89, U.S. Army Cold Reg. Res. and Eng. Lab, Hanover, N.H.Special Report 91-16, 1991.

Morin, S., Domine, F., Arnaud, L., and Picard, G.: In-situ measurement of the effective thermal conductivity of snow, Cold Regions Sci. Tech., 64, 73-80, 2010.

Riche, F. and Schneebeli, M.: Thermal conductivity of snow measured by three independent methods and anisotropy considerations, The Cryosphere, 7, 217-227, 2013.

Sturm, M., Derksen, C., Liston, G., Silis, A., Solie, D., Holmgren, J., and Huntington, H.: A reconnaissance snow survey across northwest territories and Nunavut, Canada, April 2007, Cold Regions Research and Engineering laboratory, Hanover, N.H.ERDC/CRREL TR 08-3, 1-80 pp., 2008.

Sturm, M., Holmgren, J., Konig, M., and Morris, K.: The thermal conductivity of seasonal snow, J. Glaciol., 43, 26-41, 1997.

Taillandier, A. S., Domine, F., Simpson, W. R., Sturm, M., Douglas, T. A., and Severin, K.: Evolution of the snow area index of the subarctic snowpack in central Alaska over a whole season. Consequences for the air to snow transfer of pollutants, Environ. Sci. Technol., 40, 7521-7527, 2006.

Yen, Y.-C.: Review of thermal properties of snow, ice, and sea ice, United States Army Corps of Engineers, Hanover, N.H., USACRREL Report 81-10, 1-27 pp., 1981.

Florent Domine, 24 February 2022

---

## Author Response (AR2)

**Author Response for** *"Impact of measured and simulated tundra snowpack properties on heat transfer"*, **Victoria R. Dutch et al.**

We would like to thank the editor and both reviewers for taking the time to read and comment on the original manuscript. As a consequence of these reviews, we have made three significant changes:
1. Text describing the derivation of the coefficients required to process the SMP data estimate snow density was shifted from Appendix A and is now prominent in the text of the manuscript (it was clear from the comments that Reviewer 3 did not read the Appendix).
2. The parameterization of Fourteau et al. (2021) has been added to the analysis as suggested by Reviewer 3.
3. We re-ordered the presentation and discussion so the sensitivity to different parameterizations of snow thermal conductivity is presented before the correction factor analysis.

We also provide responses with corresponding manuscript changes to all review comments, which are outlined below. For ease, reviewer comments are in black and our responses in blue. We respond to the comparatively minor revisions of Reviewer 4 before Reviewer 3.

**Reviewer 4:**
This is a solid study that makes use of new detailed high vertical resolution snow thermal conductivity measurements to assess the ability of the Community Land Model to represent the snow thermal conductivity. The authors find, somewhat unsurprisingly, that the lack of a process representation of depth hoar formation in CLM leads to much higher snowpack effective thermal conductivities than are observed at these sites. The paper includes a good discussion of paths forward, including introducing an adjustment factor and utilizing alternative snow thermal conductivity formulations. Overall, I think the authors did a thorough job responding to the comments of the prior reviewers. From my perspective this paper is suitable for publication with minor revisions suggested below.

1. I found typos throughout the revised manuscript. Many of them looked like they arose through accepting track changes wherein periods or other punctuation was lost. I suggest that the authors carefully review the manuscript to correct these typos.

Apologies for the typos in the revised manuscript, which resulted from some issues between the track changes and clean version. We have corrected these in the new version.

2. The discussion of the snow model in CLM5 is reasonable, but there is no mention of fresh snow density. The formulation of fresh snow density was modified substantially with a new temperature relationship and the addition of wind effects. In other assessment of CLM, we found that these changes resulted in higher and more realistic bulk snow densities (van Kampenhout et al., 2017, Lawrence et al. 2019), though the lack of widespread availability of snow density data makes these assessments more qualitative than quantitative. In any case, the fresh snow density parameterization should be introduced as well, I think, for the sake of completeness in the description of the snow model. I don't think the full set of equations need to be included, but a description would be helpful. The equations can be found here: https://escomp.github.io/ctsm-docs/versions/release-clm5.0/html/tech_note/Snow_Hydrology/CLM50_Tech_Note_Snow_Hydrology.html#ice-content

The following sentences have now been added to the end of the paragraph beginning on line 137:

"Changes implemented in CLM5 also include a new snow densification scheme, whereby fresh snow density is parameterised as a function of temperature and wind speed. The density of fresh snow can increase through the process of wind-driven compaction if wind speeds exceed 0.1 m$^{-1}$ (Van Kampenhout et al., 2017)."

3. In a response to a reviewer, the authors note that they are exploring what could be done in terms of changes that would improve the model at the global scale. But, there is no mention of this in the text. As an interested reader, I was very curious about what approach could be taken at global scale, since this bias correction parameter would be difficult to define globally and the Sturm parameterization only really applies for tundra snowpacks. I think it would be helpful for a reader to elaborate just a bit more on the challenges here and to note explicitly that it is under investigation in an ongoing study.

We suggest the application of the Sturm parameterisation could be based on the dominant PFT for the model grid cell. The same logic could be used for the application of the correction factor α, and we had initially implied this. However, in the revised version, we no longer recommend this because the optimal values of α depend on snow depth and hence α changes at the same location between years (and would most probably also vary between locations).

4. If I am understanding correctly (I might not be, in which case the text needs to be modified to make this clearer), the same bias correction factor is applied to all the layers in the simulated snowpack. But, the main limitation of the model is that there is not representation of depth hoar that forms at the base of the snowpack. Would it be possible or advisable to apply a correction factor to just the deepest model snow layer to try to replicate this real world behavior?

Prior to applying the correction factor to all snow layers, we tested applying the correction to just snow layers 3 & 4 (bottom two layers - shown in blue on Fig. 5). However, because these layers are not initialised until later in the winter, the soil gets too cold at the start of the winter and doesn't really recover. This was particularly evident in 2017-18 due to the shallower snowpack. Because snowpack metamorphism is a continuous process and not only enacted at an arbitrary depth threshold, it is more appropriate to bias correct all layers starting from the initial snow accumulation.

One reason I ask is that in a study that was just published, we found that winter daily temperature variability is substantially improved in CESM2, where improved is towards less temperature variability. We inferred in that study that this reduction in variability can be largely attributed to the higher snow densities and relatedly higher thermal conductivities that acted to reduce air temperature variability due to the larger effective surface heat sink. The results of your present study imply that the thermal conductivities in CLM5 are too high, thus possibly countering the result of our study on temperature variability. However, if the uppermost snow layers are still actually relatively dense and it is just the deepest snow layer that has low thermal conductivity, it's possible that this result on temperature variability is still valid. Anyway, not sure that you need to address this comment much or at all in the manuscript, though it might be interesting as a point of reference to mention this paper (it's another argument for why snow thermal conductivity matters, in addition to the soil respiration that you already mention). The paper is Simpson et al., 2022, JAMES, https://doi.org/10.1029/2021MS002880

Thanks! This is good to know. Because we did not run the model in coupled mode, we agree that this need not be mentioned in the manuscript. However, we are taking this work further into

implications for Net Ecosystem Exchange so knowledge of this new paper will be valuable for this future work.

5. The Figure 5 figure caption needs more info about what the colors mean. I was able to infer that they indicated the snow layers (nice way to look at the evolution of the snowpack layers in CLM!), but I guess some readers may not be able to recognize that immediately.

Added the following to the caption. It now reads:

"Figure 5: Simulated snow layers and their properties for winter 2017-18; a) simulated snow layer thicknesses, b) snow layer densities, c) snow layer thermal conductivities; d) to f) as before but for winter 2018-19. Each colour represents a different computational snow layer."

**Reviewer 3:**

*General Comments*

These general comments were written after the subsequent specific comments, I apologize for the few repetitions.

This paper in fact reports two somewhat independent topics. The first one is the determination of vertical thermal conductivity profiles of snow near TVC using density profiles determined from SMP profiling. Density is derived from SMP data using an algorithm developed from snow on sea ice, if my understanding is correct.

We draw your attention to Appendix A, which clearly states new coefficients specific to tundra snow have been derived by applying the King et al (2020) methodology. The sentence pointing the reader to the appendix has been reordered slightly to make this more explicitly obvious, it now reads: "Bespoke coefficients for tundra snowpacks were calculated based on the methodology of King et al. (2020) to derive high vertical resolution snow density profiles from the SMP force profiles (see Appendix A for detailed methodology)." However, the reviewer may be interested to know that the values from snow on sea ice (King et al., 2020) and those we derived from snowpit density profiles at TVC were relatively similar (both are given in Table A1, alongside the original coefficients of Proksch (2015) derived using an earlier version of the SMP).

The second one is to compare these thermal conductivity profiles to those simulated by CLM. As expected from any snow model used in the Arctic, CLM does not perform well. It simulates much too cold temperatures and the authors discuss various adjustements to make the model better fit the data.

The experimental part is interesting and there is some novelty to it. There are however a number of weaknesses. First, the thermal conductivity is derived from density using various equations that do not agree with each other. Second, while I am willing to believe that the algorithm used to convert SMP data to density is good, it seems to have been validated for snow on sea ice, which is somewhat different from snow at TVC where vegetation impacts snow properties.

As described in Appendix A, the estimation of snow density from the SMP measurements was fully developed using only the tundra snow measurements. Coefficients were derived from Jan 2019 and March 2019 TVC field campaigns. We have now made these methods more prominent in the main body of the text by shifting some of the text from the appendix:

"Bespoke coefficients for tundra snowpacks were calculated based on the methodology of King et al. (2020) to derive high vertical resolution snow density profiles from the SMP force profiles (see Appendix A for detailed methodology). Briefly, a K-folds recalibration was used to derive new coefficients (Table A1) from 36 co-located snowpits and SMP profiles across the TVC catchment. These coefficients were then applied to all 1050 SMP force profiles from the 3 campaigns over a 2.5mm rolling window to give recalibrated density profiles. These density profiles were then used to approximate profiles of thermal conductivity using the $K_{eff}$ relationships derived by Sturm et al. (1997), Calonne et al. (2011), Jordan (1991), and Fourteau et al. (2021), denoted $K_{eff-Sturm}$, $K_{eff-Calonne}$, $K_{eff-Jordan}$ and $K_{eff-Fourteau}$ respectively."

Third, in the discussion, the authors mention an artefact where the penetrating SMP rod may have damaged the fragile depth hoar, causing an artefact with lower intermediate density and increased

lower density. This third issue is very important and must be mentioned in methods so that the reader has it in mind while evaluating the experimental part.

This is mentioned in detail in Appendix A as well as in the discussion. We now repeat some of this in the main body of the methods, as follows:
"Sources of uncertainty in the SMP measurements include interactions with vegetation within the snowpack and collapse of the depth hoar layer during measurement; an experienced SMP user can easily identify and remove profiles which are affected by these issues. A positive bias in derived depth hoar density occurs because of large distances between snow grain failures (see Appendix A and King et al., 2020b for more details)."

I detail this point below. So, while SMP clearly allows obtaining much more data with high vertical resolution, its limitations and potential artefacts must be stressed, with the possible conclusion that it is not (very) suitable for snowpacks with very fragile basal layers. Or that the wind slab must be removed to measure the depth hoar? In any case, I feel the SMP data deserves publication if the caveats are clearly stated.

We appreciate that uncertainties may be introduced to SMP measurements by collapse of the depth hoar layer. As mentioned above, we note this source of uncertainty in the text. With respect to mitigating this uncertainty, profiles with obvious depth hoar collapse or depression in the wind slab during measurement were removed. Experiments with removing the wind slab layer would likely result in disturbance to the remaining depth hoar, so it is not obvious how this approach would help. In addition, with the comments of this review in mind, stratigraphy and density profiles with 100 cm$^3$ box cutters made at TVC in late March 2022 were closely examined and the measurements showed an increase in density close to the snow-ground interface. This section of depth hoar was much more consolidated, most likely due to more moisture being available in the upper soil/veg layers at the onset of snow accumulation. In short, we now have more confidence in the increase in density close to the snow-ground interface shown by SMP profiles than when we wrote this manuscript and so the caveats do not need to be further affirmed in addition to what is already presented in this manuscript.

Another issue with the derivation of thermal conductivity (TC) profiles from SMP data is the choice of the density-TC relationship. First, the authors must remain aware that this is far from a bijective relationship and that there is enormous scatter in this relationship. For example, in (Sturm et al., 1997) there is sometimes a factor of 5 variation in TC for a given density value. The use of such an equation may therefore bear some implicit very high uncertainty and error. Second, there are large differences between the various relationships used, so which one is correct, if there is one, and what is the error induced by these relationships?

In our view, none of the density-TC relationships are correct because they all presume snow thermal conductivity to purely be a function of density, whereas we know other factors such as grain shape and temperature are also important as we acknowledge in the text. We devised an approximate snow thermal conductivity from the SMP using the density relationships to give a first-order approximation of the snowpack thermal conductivity, which is consistent with many other studies. Showing multiple relationships and the interquartile range of the values produced by each relationship gives an idea of the uncertainty across these approximations.

I would like to comment on the various relationships used. The equation of (Jordan, 1991) is based on that of (Yen, 1981) which is in fact a compilation of previous works using a variety of methods. (Sturm et al., 1997) use only the needle probe method, but there are doubts on its reliability (Riche and Schneebeli, 2013) and this is not just due to anisotropy issue as stated by the authors. (Fourteau

et al., 2022) have very recently made a detailed theoretical and experimental study of the needle probe method and demonstrated that the aproximate equation used by (Sturm et al., 1997) to derive thermal conductivity from heated needle probe data is not correct, leading to a negative artifact. Briefly, it is not valid for the short heating times used as some neglected terms in the full equations are in fact not negligible. (Fourteau et al., 2022) proposed an algoritm to correct existing needle probe data. In any case, I think the equation of (Sturm et al., 1997) is incorrect, and the fact that it produces values lower that the other parameterizations confirms this negative artefact. I very strongly recommend to stop using this equation. The equation of (Calonne et al., 2011) is based calculations from tomographic images. It is rigorous, however, it minimizes water vapor effects because is postulates slow surface kinetics. In other words the accommodation coefficient of water vapor on ice takes here a low value, so that latent heat fluxes are minimized and probably underestimated. More recently (Fourteau et al., 2021) used a similar approach but postulated fast surface kinetics, leading to slightly higher values than (Calonne et al., 2011). In conclusion, the equation of (Sturm et al., 1997) is incorrect and should just not be used. The equation of (Jordan, 1991) is based on ancient measurements using a variety of poorly documented techniques and I feel it has limited reliability. The equation of (Calonne et al., 2011) is arguably correct but makes the debatable postulate that surface kinetics is slow, while (Fourteau et al., 2021) make the more reasonable (in my opinion, but I am a coauthor of that paper) postulate of fast surface kinetics. My biased recommendation is therefore to use the equation of (Fourteau et al., 2021) and in any case not to use the equation of (Sturm et al., 1997). I recommend not to use different equations as an error compensation trick in CLM.

We thank the reviewer for their thoughts on the relative merits of each relationship. We have added the relationship from Fourteau et al. (2021), which was not yet published when we were preparing the original manuscript.  We make no judgement on the accuracy of any particular functions for this environment, rather we include them to quantify the spread between the relationships and not as any form of error compensation. It is interesting to note there are only small differences in the results produced from the Calonne, Jordan, and Fourteau parameterisations, both when applied to the SMP measurements and when used within CLM. Consequently, sensitivity to different model parameterizations is now considered in the manuscript first, with the correction factor analysis being a subsequent argument.

In fact, I am not too thrilled by the modeling part. Essentially this just says that a simple snow routine from a more complex model just does not work in the Arctic. It is interesting to confirm that the CLM snow scheme is deficient, but I do not think all the error compensation tricks used by the authors, and the lengthy discussion on parameter adjustments, have much interest. Figure 7 shows that for the first year, α =0.4 must be used while for the second year α=0.6 is better. So, the adjustment parameter changes from year to year, and is probably different for other sites, so that CLM has no predictive value when it comes to Arctic snow and ground temperature. We already know, I as well as all the authors, that even the currently most sophisticated snow model do not work in the Arctic, so no one expects the simpler CLM scheme to perform any better. The authors just need to show Figure 7, demonstrating the lack of predictive value of CLM and therefore its much reduced interest for predicting snow properties and soil temperature in the Arctic. They then could just discuss that implementing the missing process, upward vapor transfer, is too complex, and that other approaches they wish to propose must be envisaged. I Therefore think Figures 1, 2, 3, and 7 are interesting. I strongly recommend that the authors consider removing the other Figures, or at least most of them, and reduce the modeling text by at least 50%, probably more. Please also consider my comment of Figure 5 below.

Given the widespread usage of CLM, particularly for high-profile climate projections, this finding is an important one and it is critical to bring this to the attention of people outside of the snow

microphysics community. This viewpoint is echoed by Reviewer #4 who has significant experience in land surface modelling and is a developer of CLM.  We have chosen this journal as a mechanism to bring these problems to the attention of the wider modelling community, particularly those who might be able to fix it.  We were indeed not surprised that CLM performs poorly in its representation of the vertical properties of Arctic snowpacks. However, the implications of snowpack simulations on soil temperatures have broader consequences in earth system models. For example, underestimation of soil temperatures impacts simulations of soil respiration with implications for pan Arctic carbon fluxes (e.g. Natali et al. (2019)).

In addition, both reviewers from the first round of the review process also had positive things to say, with reviewer #2 describing it as "very relevant" and reviewer #1 saying that it is "only using such work presented in this paper that we could be able to better understand the soil-snow-atmosphere feedback in the Arctic." So, while reviewer #3 may not be thrilled by this modelling analysis, we are hopeful that the detailed analysis and implications will be more appreciated by the wider community.

To help better engage that community, in the revised version of the manuscript, the modelling text has been condensed and figure 8 has been removed. We have re-emphasised the variability in the ideal value of the correction factor, and use this as a tool to discuss the limitations of the model rather than advocate for the application of it as a bias correction. We also discuss the application of different parameterisations of snow thermal conductivity to show the sensitivity of snow:soil relationships in CLM.

Figures 4 and 6 show that snow thermal conductivity can be approximated from the SMP at a very high vertical resolution, which we feel is novel and exciting. Figure 5 aides in the understanding of the numerical manner in which CLM builds a snowpack and the deficiencies that result from this. Reviewer #4 also thought this was an effective way to visualise the development of the snowpack.

Another general comment is that there is a serious lack of attention to detail in the writing and presentation. This is surprising given that "All authors were involved in reviewing and editing prior to submission" and that the authors include a large number of high-profile esteemed and highly respected senior researchers. For example (just one, and I will not edit for typos, the authors can do it), all of these authors think it is fine to write "Snow has a low thermal conductivity, typically in the range $0.01 - 0.7$ $Wm_{-2} K_{-1}$". So snow can have a thermal conductivity less than half that of air? And for brevity I will refrain from commenting the 0.7 value. In any case, I will do my best to write a hopefully constructive review. Let the authors do their best to write a paper with attention to detail.

For brevity, we will comment that the range of thermal conductivities of snow was taken from Gouttevin et al. (2018): "Snow has a low thermal conductivity ($K_{eff}$), ranging from 0.01 to 0.7$Wm^{-1} K^{-1}$ depending on microstructure, density, and wetness, and it therefore insulates the underlying ground during the cold season." The Cryosphere, 12, 3693–3717, 2018.

Finally, it was a collective oversight of the entire author team to fail to spot some typos resulting from the acceptance of track changes during revision. However, the patronising and dismissive language accompanying this admonishment is unnecessary and undermines the scientific training of the Early Career Researcher entrusted with leading this manuscript. The open review process of The Cryosphere correctly allows for full transparency within our community. Respectful treatment of ECRs is a reasonable expectation.

*Specific comments*

Line 44. I am not sure what the authors mean by "indurated depth hoar". They seem to have a definition different from mine (Domine et al., 2016b) and from Sturm's (Sturm et al., 2008). Sturm and I have the same definition, since Sturm introduced me to indurated depth hoar on the Alaska north slope in 2004. I would think the authors would have a similar definition since Chris Derksen has done much Arctic snow field work with Sturm. However their stratigraphy does not show any indurated depth hoar, but faceted crystals. This is very confusing. I would also think the lower layer, with densities reaching 300 kg m-3, would often be actual indurated depth hoar, possibly formed from melt-freeze layers. In any case, I suggest the authors realize that the classification of (Fierz et al., 2009) was made by avalanche experts for avalanche motivations. It is almost exclusively based on observations in Alpine snow and is largely inadapted to Arctic snow. I discussed this with Charles Fierz and he had never seen indurated depth hoar. I think Arctic snow researchers should use symbols adapted to their problem. I have proposed symbols for indurated depth hoar and indurated faceted crystals (stage prior to indurated depth hoar) in (Domine et al., 2016b) and (Domine et al., 2018) (already cited by the authors) and in other papers. Why not popularize these symbols, adapted to Arctic snow, which by the way is much more important area-wise than alpine snow ? It would spare us these inevitable inconsistencies between text and Figures.

We use indurated hoar as identified as "wind slab to depth hoar" layer as per Sturm et al. 2008 and as described in Domine et al. 2016. To avoid unwanted confusion we have now cited Sturm et al. (2008) in our introduction:
"Between these two layers, an indurated hoar layer may also be formed (Sturm et al., 2008), where the lower part of the wind slab takes on some of the microstructural properties of depth hoar (e.g faceted grains) while maintaining the density and hardness of a wind slab (Derksen et al., 2009)."

Symbols for indurated depth hoar do not exist in the international classification of Fierz et al. (2009). We have used a symbol for facets crystals as the slab layer was turning towards depth hoar. However, if they can be made available by the reviewer in a publicly available font file (.ttf), we will add them to Figure 2 which shows symbology in an indicative manner to help the reader interpret stratigraphy. The alternative is to remove the symbol for faceted crystals from Figure 2.

Line 150. What is hsl?

$h_{sl}$ is the height of the snow layer. This is given in the explanation sentence below eq. 1 (what was L154).

Line 185-190 are not necessary. These are very well-known considerations. In general section 3.1 can be greatly condensed.

Section 3.1 has been condensed.

Figure 2: Faceted crystals? Columnar DH? Please clarify symbols and make them consistent with earlier parts of paper.

Figure 2 shows grain type symbology (from Fierz et al. 2009) alongside changes in density, in an indicative manner to help the reader interpret stratigraphy. If a further symbol depicting indurated depth hoar (following the slab to hoar interpretation of Sturm et al. 2008) is desired, we will apply it if the reviewer can provide it in a publicly available font file (.ttf) – see comment above regarding either removal of the facets symbol or addition of a new symbol.

Plus, again, Fierz 2009 inadapted to Arctic snow. How about showing fall 2018 data? Fall data are in general scarce amnd therefore valuable.

Unfortunately, due to instrument malfunction SMP data are not available for the November 2018 field campaign. Pit data from November 2018 are shown in Figure 3.

Table 2 lacks detail and does not correspond with stratigraphy of Figure 2. There seems to be a dense basal layer, perhaps indurated depth hoar, in the lower 10-20%. Then the next 20-60% seem to have a homogeneous low density typical of columnar depth hoar. Therefore, these 2 layers should be separated in Table 2. Just having one DH layer is not consistent with data.

We feel there is a balance to strike here. An exciting reason to use SMP measurements is that traditional binary thresholds for wind slab / depth hoar or wind slab / indurated depth hoar / depth hoar do not capture the transitional nature of snowpack properties in a vertical profile. This is arguably a significant advance on traditional snowpit stratigraphy. However, pragmatic classification of snow layers will always have a role in model evaluation, hence we feel that the three-part (surface snow, wind slab, depth hoar) classification in Table 2 provides useful information to readers, especially those to whom land surface modelling is a focus.

Line 214. Please descrive indurated DH. Written as faceted crystals in Figure 2.

See previous responses to comments on Figure 2 which directly address this issue.

Line 221. Ice lenses not shown in stratigraphy.

Ice lenses were only observed in a small number of pits, whereas figure 2 shows an amalgamation (median and interquartile range) of ~400 SMP profiles. Adding ice lens symbols to the indicative symbology of Figure 2 would not aid in conveying the properties of the total distribution of density measurements.

Line 235. Please check grammar
Done.

Line 257. Please be consistent with snow layers. Figure 2c and d mention 5 layers. Here just 3.
Our response here follows on from the comment on Table 2. The transitional element of SMP derived density profiles rather than traditional binary layer identification or computationally modelled layers, means that measurements made at the sub-millimetre scale may not directly translate. Simplification of layers in the way the arguments are presented, especially when discussing modelled layering, means this perceived discrepancy in consistency of the number of snow layer consistency is valid.

Line 265. 0.344 is 4 times as large as 0.08, not 3.

The original sentence was to state that snow thermal conductivities from CLM (0.344) are ~3 times larger than those from the needleprobe (0.08) *or the SMP (0.11)*. This has now been changed to "at least 3 times" for clarity.

Sturm is not an approximation, but an equation, or a parameterization.

The word approximation has been changed to parameterization.

Line 268. Most of Sturm's measurements were in fact in the Boreal forest of interior Alaska, where the mostly depth hoar snowpack layers have very low thermal conductivities. Many measurements were also from tundra snow but what the authors say is incorrect.

Changed to refer to "the Alaskan Arctic".

Line 281. Probably unnecessary explanation.

Removed.

Figure 5. The graphs may be easier to visualize if the grey and black colors were swapped.

Contrast between the colours of the different layers alternates in order to make the individual layers easier to distinguish, particularly for readers who are colour blind. Putting black and navy blue layers next to each other would make the figure harder to read.

In any case, I am not sure about the utility of panels b-d and in fact I am surprised by how CLM seems to work. The origin seems to be the top rather than the base of the snowpack, even though the snowpack forms from the bottom up, as the authors know. Following density from the top then does not really correspond to anything physical, as a given snow layer is not monitored. I do not have time to get into the methods and architecture of CLM, which is totally unknown to me, but it seems very strange. Since it does not seem to take into account actual processes, it seems that having such a model fit data will result in mandatory adjustments on a case by case basis, with no hope of ever having any predictive value. From what little I understand, I therefore wonder whether there is actually any hope of ever getting any reliable snow simulations from CLM. I'll be more than happy to be proven wrong.

The process by which CLM builds a snowpack is entirely computational and does not reflect the physical processes which create an Arctic snowpack. However, this does not mean that any adjustment can only be made on a case-by-case basis or that the model has "no hope of ever having any predictive value". Improvements to the properties of this simplified snowpack structure can still provide an improvement to the simulation of snow insulation and soil temperatures for example, such as when reducing the simulated thermal conductivities. Changes to this simple snow model, such as through the substitution of snow thermal conductivity parameterisation provide further improvement to the simulated subnivean conditions.

Line 293. Why just mention 2018-2019? Is not it interesting to realize that in 2017-2018, α =0.4 works best?

The following sentence has been added:
"However, a smaller value of α was required for best model performance in 2017-18, with an α of 0.4 giving the lowest RMSE of 1.6°C and highest SHTM of 0.986"

A single value cannot simulate both years. And therefore, expectedly, different thermal conductivity parameterizations have to be used for each year. This may be stressed.

The ideal value of the correction factor changes between years, and more emphasis has now been placed on this in the text.

Line 296. I do not understand (0.3 ≥3≥ α ≥ 0.5555)

Thanks for flagging this. This is a typo as a result of merging different sets of tracked changes and has now been fixed to read (0.3 ≥ α ≥ 0.55).

Line 316. Sturm's parameterisation works best, but that parameterization is wrong. This is just an error compensation game. The model is wrong, and this is compensated by a wrong thermal conductivity parameterization.

All models are imperfect. The value of these simulations comes from what they tell us about the dynamic relationship between snow and soil, and how this is influenced by both the application of alternative snow thermal conductivity parameterisations and the use of the correction factor α. All parameterisations are by necessity a simplification, and as you previously mentioned, none of them are perfect. We feel using the largest reduction in RMSE, common in snow modelling evaluation, to guide parameterization choice is a valid approach.

And the compensation is different for each year, meaning that the model has no predictive value.

The agreement between simulated and observed soil temperatures changes within and between winters. This is the case regardless of the thermal conductivity parameterisation used. However, we dispute the assertion that this means the model has "no predictive value" because it does provide insight into the potential for bias correction. The ideal value of α does change between years, and more attention is now brought to this in the text. As detailed in figure 7c and Table 1, CLM considerably underestimates the depth of the snowpack in 2017 − 18. This is most likely due to uncertainties in observed precipitation values used to force the model (as mentioned in the methods), and we believe this error in snowpack depth is the most likely reason that the model gives better soil temperature simulations (and requires a smaller correction) in 2018 − 19. Consequently, in the overall narrative of manuscript we changed the sequence of the results to first show the sensitivity of CLM to the various thermal conductivity parameterizations, and then applied the correction factor as an illustration of bias correction.

Lines 326-328. This comparison is very misleading. The authors compare their indirect estimation of thermal conductivity based on an indirect estimation of density to actual measurements of snow thermal conductivity. Furthermore, the measurements of (Domine et al., 2015; Domine et al., 2016b) and of (Morin et al., 2010) are continuous season-long time series, so that the focus of those papers are on time-variations, while the authors'work is on height variations.

We have no intent to mislead. Rather, as stated in the text, it is a novel approach to use very high vertical resolution estimates of density from SMP measurements to try and understand changes in the snowpack thermal conductivity. From our perspective, this leverages and promotes knowledge from the classical thermal conductivity studies we cite, a larger mistake would be to ignore this and not cite it.

By the way, vertical profiles of thermal conductivity measured by (Gouttevin et al., 2018) and by (Domine et al., 2012; Domine et al., 2016a) is closer to 5 cm resolution than to 10 cm.

Changed to read "~ 5 − 10 cm".

Lines 330-344. This discussion is interesting but I think it should already be stated in the methods section. I have been wondering about this high density basal layer, thinking it may reflect rain-on-snow in the fall but only now do I realise it is probably just an artifact! I have been misled in my understanding of the data all along! So please shift this up. And by the way, why did not the authors

perform SMP measurements with the wind slab remove to test for the actual impact of this artefact, since they must have been aware of it while making the measurements.

This was stated in the appendix as well as being re-iterated in the discussion. The following text has also been added to the methods section:
"Sources of uncertainty in the SMP measurements include interactions with vegetation within the snowpack and collapse of the depth hoar layer during measurement; an experienced SMP user can easily identify and remove profiles which are affected by these issues. A positive bias in derived depth hoar density occurs because of large distances between snow grain failures (see Appendix A and King et al., 2020b for more details)."

Please also see our response above to a similar comment where very recent measurements at TVC in late March 2022 suggest this higher density depth hoar near the soil-snow interface is not an artefact.

Lines 345-352. Continuous vertical profiling is indeed very nice and is clearly a significant improvement over discrete layer sampling. However, is SMP suited to Arctic snowpacks, with a basal depth hoar layer that can be extremely fragile and collapse at the slightest touch? (see details in (Domine et al., 2016b). The authors seem aware of this problem, and this clearly limits the interest of SMP for some Arctic snowpacks. Not all, I agree, since very windy areas such as Barrow and polar deserts seldom have very fragile depth hoar. This should also be discussed. Furthermore, SMP is blind sampling, since a snowpit is not dug in most cases as this would cancel the benefit of the technique. Therefore, artefacts due to soft layers would be undetected. In conclusion, while I do see the benefit of SMP in some cases, the authors may wonder whether a low resolution reliable manual density profile is better than a high resolution profile with potential and unverifiable artefacts.

We have never advocated for not making manual snowpit measurements. In fact, some manual density profiles from snow pits are required to derive coefficients to convert SMP force measurements to density. Snowpits are also used to give an idea of what features might be likely within the SMP profiles, and which are more likely to be artefacts. Our sampling protocol required taking SMP profiles at pit locations as well as across the landscape, with the nearest neighbour SMP to each pit used to derive the Force: Density relationship. The necessity of both snowpit and SMP measurements to derive recalibration coefficients is detailed in Appendix A and is now re-iterated within the methods section as follows:
"Use of the SMP allows for a large increase in both the number of sites and the vertical resolution at each site compared to traditional snowpits, but some coincident snowpit measurements are still required to derive the coefficients to estimate snow density."

Line 355. Alpine snow does have a density profile as simulated by CLM, but not the taiga snow, which by the way is not discussed by the references cited. In the boreal forest (for some reason, I refrain from using Russian words these days…?) the profile may be as in CLM at the beginning of the season but by the end of winter it is either flat or with lower basal density because of the upward vapor flux. See e.g. (Taillandier et al., 2006).

The reference to taiga snow has been removed.

Line 361-362. "such as the snowpack vapour kinetics necessary to form depth hoar." The term "vapor kinetics" is unclear. Use flux, and more accurately upward flux.

The word "kinetics" has been replaced with "flux".

**References:**

Derksen, C., Sturm, M., Holmgren, J., Liston, G. E., Huntington, H., Silis, A., and Solie, D.: Northwest Territories and Nunavut Snow Characteristics from a Subarctic Traverse: Implications for Passive Microwave Remote Sensing, Journal of Hydrometeorology, 10, 448-463, 10.1175/2008jhm1074.1, 2009.

Fourteau, K., Domine, F., and Hagenmuller, P.: Impact of water vapor diffusion and latent heat on the effective thermal conductivity of snow, The Cryosphere, 15, 2739-2755, 10.5194/tc-15-2739-2021, 2021.

Gouttevin, I., Langer, M., Löwe, H., Boike, J., Proksch, M., and Schneebeli, M.: Observation and modelling of snow at a polygonal tundra permafrost site: spatial variability and thermal implications, The Cryosphere, 12, 3693-3717, 10.5194/tc-12-3693-2018, 2018.

Natali, S. M., Watts, J. D., Rogers, B. M., Potter, S., Ludwig, S. M., Selbmann, A.-K., Sullivan, P. F., Abbott, B. W., Arndt, K. A., Birch, L., Björkman, M. P., Bloom, A. A., Celis, G., Christensen, T. R., Christiansen, C. T., Commane, R., Cooper, E. J., Crill, P., Czimczik, C., Davydov, S., Du, J., Egan, J. E., Elberling, B., Euskirchen, E. S., Friborg, T., Genet, H., Göckede, M., Goodrich, J. P., Grogan, P., Helbig, M., Jafarov, E. E., Jastrow, J. D., Kalhori, A. A. M., Kim, Y., Kimball, J. S., Kutzbach, L., Lara, M. J., Larsen, K. S., Lee, B.-Y., Liu, Z., Loranty, M. M., Lund, M., Lupascu, M., Madani, N., Malhotra, A., Matamala, R., McFarland, J., McGuire, A. D., Michelsen, A., Minions, C., Oechel, W. C., Olefeldt, D., Parmentier, F.-J. W., Pirk, N., Poulter, B., Quinton, W., Rezanezhad, F., Risk, D., Sachs, T., Schaefer, K., Schmidt, N. M., Schuur, E. A. G., Semenchuk, P. R., Shaver, G., Sonnentag, O., Starr, G., Treat, C. C., Waldrop, M. P., Wang, Y., Welker, J., Wille, C., Xu, X., Zhang, Z., Zhuang, Q., and Zona, D.: Large loss of $CO_2$ in winter observed across the northern permafrost region, Nature Climate Change, 9, 852-857, 10.1038/s41558-019-0592-8, 2019.

Sturm, M., Derksen, C., Liston, G., Silis, A., Solie, D., Holmgren, J., and Huntington, H.: A reconnaissance snow survey across northwest territories and Nunavut, Canada, April 2007, Cold Regions Research and Engineering laboratory, Hanover, New Hampshire, 1–80, 2008.

---

## Author Response (AR3)

Thank you for accepting our paper for publication in The Cryosphere! We thank the reviewers for their decision and the time they have taken to review this paper. In light of the comments by Reviewer 4 (in black – only the relevant parts of the reviewer comments are shown), we have made some minor alternations (in blue) to the text in the discussion section:

However, one thing that occurred to me is that the authors note that the simulated snow depth is biased low in the CLM simulations. This low bias makes it difficult to know what emphasis should be placed on the simulated soil temperature biases in the control and modified versions of the model. It may be beyond the scope of this study, but did the authors consider using this low snow depth bias to try to correct the snowfall rates at these sites so as to drive a more realistic seasonal and end of season snowpack depth.

After the following text in the original,

"We note that issues in simulating the initial accumulation of the snowpack are likely linked to uncertainties in the forcing data caused by measurement limitations surrounding the use of precipitation gauges in tundra environments (Smith, 2008; Watson et al., 2008; Pan et al., 2016)."

We have added this text to the discussion (on line 425):

"However, attempting to correct for snow depth errors through adjustment to the precipitation forcing beyond the corrections outlined in Pan et al. (2016) is not advisable due to high variability of snow depth (Fig 2a) over short spatial scales (metres to tens of metres). Additionally, Fig 7c suggests that the timing of the snow onset is more important in determining the soil temperature than the absolute snow depth error, as in 2018-19 soil temperatures simulated using the Sturm parameterisation are closer to observations than in the previous year, despite an absolute snow depth error of up to 0.2m."

If the snowpack depth was represented more realistically, then the comparison of modeled and observed soil temperatures would be more meaningful. I wonder if the relative impact of the different tested parameterizations would be different under a low snow depth bias. If such a simulation is not easy to do, then I would at least suggest a better discussion of the potential interpretation error that the low snow depth bias could impart.

Additional simulations at this point are beyond the scope of this study. However, we added the following text to the discussion (on line 410):

"A similar bias compensation effect could apply for the use of alternative parameterisations of snow thermal conductivity. If snow depth bias was consistently positive, we suspect that the Calonne, Fourteau and Jordan parameterisations would likely compensate for an overthickened snowpack through increased thermal conductivity. However, under a negative snow depth bias the Sturm parameterisation remains more suitable; although the absolute magnitude of the improvement in soil temperatures using the Sturm parameterisation was lower when the snow depth bias was greater in 2017-18, the relative order of impact of the different parameterisations remained the same."